# Naturally occurring dinactin targets *cpsA* protein and kills *Mycobacterium tuberculosis* by disrupting the proton motive force

Gaoyan Wang[1,2,3,4], Wenqi Dong[1,2,3,4], Yajuan Bai[1,2,3,4], Yuxin Li[1,2,3,4], Hao Lu[1,2,3,4], Wenjia Lu[1,2,3,4], Chenchen Wang[1,2,3,4], Jia Tang[1], Pei Li[5], Rui Wang[6], Xiangru Wang ®[1,2,3,4], Huanchun Chen[1,2,3,4] & Chen Tan ®[1,2,3,4] ✉

Tuberculosis, especially drug-resistant tuberculosis remains a global threat, and new drugs are desperately needed to combat the spread of multidrug-resistant *Mycobacterium tuberculosis*. Here we describe a natural macrotetrolide dinactin with anti-tuberculosis activity against susceptive and non-replicating *Mycobacterium tuberculosis*. Dinactin can also synergistically enhance the anti-tuberculosis effect of rifampicin and isoniazid against drug-resistant strains.Furthermore, dinactin exhibited excellently antituberculosis effect in macrophage and *Galleria mellonella* models. Since the ionophore properties of dinactin, it not only enhanced cations transport and altered membrane permeability but also caused the dissipation of proton motive force and metabolic perturbations. Finally, through the selection of spontaneous resistant mutants and whole genome sequencing, non-synonymous single nucleotide polymorphisms were successfully identified in the *cpsA* gene of the LytR-Cps2A-Psr family. The dinactin-resistant mutants exhibited decreased in vitro drug sensitivity to dinactin without cross-resistance to first-line antituberculosis drugs. Genetic studies and molecular biology assays have subsequently confirmed *cpsA* as one of the potential targets for dinactin's anti-tuberculosis activity. Collectively, these data indicate that dinactin could be a promising candidate for treating tuberculosis.

Tuberculosis is an infectious disease caused by the pathogenic *Mycobacterium tuberculosis* (*M. tuberculosis*), which still is one of the greatest threats to global public health, with tuberculosis causing millions of deaths every year[1]. The current standard treatment course for drug-susceptible tuberculosis relies on a combination of four first-line antituberculosis antibiotics for 6 months[2]. However, the emergence of multidrug-resistant (MDR) and extensively drug-resistant (XDR) strains has caused great difficulty in tuberculosis therapy, for which the standard regimen is ineffective against these resistance mutants. Meanwhile, the new antituberculosis drug discovery is relatively slow in progress. Although two new antituberculosis drugs, bedaquiline, and delamanid, have been approved to treat drug-

resistant tuberculosis, their reported toxicity has limited their widespread use[3]. To effectively combat the growing challenge of drug resistance, researchers must work together to find novel compounds that can shorten and improve therapy duration.

The discovery of streptomycin, a *Streptomyces sp*-derived antibiotic, ushered in an era of natural products (bacteria-derived) as the drug treatment of tuberculosis. This also triggered decades of research in developing antituberculosis drugs, laying the foundation for the standard regimen therapies for drug-sensitive tuberculosis[4,5]. Natural products continue to be a great resource for the discovery of antituberculosis drugs, and they have provided a wide range of structural diversity in bioactive molecules with

[1]State Key Laboratory of Agricultural Microbiology, College of Veterinary Medicine, Huazhong Agricultural University, Wuhan, China. [2]Hubei Hongshan Laboratory, Wuhan, China. [3]Key Laboratory of Preventive Veterinary Medicine in Hubei Province, Wuhan, China. [4]International Research Center for Animal Disease, Ministry of Science and Technology of the People's Republic of China, Wuhan, China. [5]National Clinical Research Center for Infectious Diseases, Guangdong Provincial Clinical Research Center for Tuberculosis, Shenzhen Clinical Research Center for Tuberculosis, The Third People's Hospital of Shenzhen, Shenzhen, China. [6]Department of Experimental Animal Center, West China Hospital, Sichuan University, Chengdu, China. ✉e-mail: tanchen@mail.hzau.edu.cn

good activity against *M. tuberculosis*[6]. Regrettably, large pharmaceutical companies' interest in developing antibacterial drugs has declined, resulting in a significant amount of research on promising natural products being shelved. However, natural products research is being revitalized due to the urgent need to find novel antibiotics and the advanced genome and meta-genomic sequencing technologies becoming available[5]. Notably, the successful application of these technologies enables researchers to synthesize the semisynthetic spectinamide and sansanmycins that are active against drug-resistant and -sensitive *M. tuberculosis* strains and identify *InhA* as the mycobacterial target of pyridomycin[7–9]. Lead compounds identified strategies that use phenotypic screening of chemical or natural product libraries for whole-cell activity against *M. tuberculosis* have played an important role in enriching the TB drug pipeline. Furthermore, these tactics also help to identify novel targets and explore new mechanisms of action in *M. tuberculosis*[10,11]. The successful precedent was the discovery of bedaquiline with a special mechanism of action, targeting the energy metabolism pathway of *M. tuberculosis* and benzothiazines (BTZ043), which inhibit cell-wall synthesis[12,13]. Although it is difficult to explore the mode of action of lead compounds derived from whole-cell screening, combining the post-genomic technologies may overcome this deficiency[11,14].

Here, we performed a whole-cell screening and found dinactin had bactericidal activity against *M. tuberculosis*. Dinactin is a macrotetrolide antibiotic with antimicrobial activity that forms a large tetra-lactone ring by inserting tetrahydrofuran moieties[15]. Dinactin is produced by the actinomycete *Streptomyces sp.* and the biosynthesis of dinactin was first studied in 2016 when the involvement of the *nonS* gene encoding for the closure of the tetrahydrofuran ring in dinactin biosynthesis was proposed[16]. Although dinactin has been described as a lead compound with significant activity against both drug-susceptible and –resistant *M. tuberculosis*[17], the action of dinactin against *M. tuberculosis* has not yet been clarified, and its potential effect as an antituberculosis drug has not been evaluated. In the present study, we elucidate that dinactin inhibits *M. tuberculosis* in vitro and in vivo. Subsequently, we used a combination of approaches involving microbiology, molecular biology, and genetic validation to clarify the mechanisms of dinactin. Our findings reveal that dinactin acts as a natural *cpsA* inhibitor as well as an ionophore which disrupts the normal energy metabolism of *M. tuberculosis* that may be further optimized to be a therapeutic agent for tuberculosis.

## Results

### Discovery of a natural compound against *M. tuberculosis*
To identify natural compounds with activity against *M. tuberculosis*, we applied the whole-cell phenotypic screening assay for preliminary screening of a natural products library, containing over 6000 structurally diverse extracts from plants and microorganisms. This phenotypic screening assay aims to identify natural compounds that target any essential pathway under in vitro conditions. The library afforded two compounds at a single concentration of 10 μM with significant growth inhibition against BCG (Supplementary Fig. 1). Further validation of the result from the primary screen, the hit compounds antibacterial ability against *M. tuberculosis* H37Rv in vitro was also tested by using REMA assay and two compounds, dinactin (Fig. 1A) and ilexsaponin A1 (Fig. 1B). Of the two hit compounds, only dinactin exhibited potent inhibitory activity against *M. tuberculosis* H37Rv with an MIC of 0.5 μg/ml (Fig. 1C), and its anti-tubercular activity was consistent with previous reports[17]. Previous studies have shown that non-actin, dinactin, monactin, and trinactin were homologs and belong to the macrotetrolides family, and that all compounds were isolated from the fermentation products of *Streptomyces species*[15,18]. The main structure of nactin was composed of alternating tetrahydrofuran rings and ester linkages. The difference between these antibiotics is the preceding one-by-one methyl group[19]. This structural difference may map to their biological activity. Thus, we also evaluated the antituberculosis activity of these homologs. Compared to the other three compounds, dinactin exhibited better activity against *M. tuberculosis* H37Rv, and it also showed the most selective index against *M. tuberculosis* (Table 1). Besides, the hemolysis assay

of these macrotetrolides against ovine red blood cells was measured, and no lysis of blood cells was observed (Fig. 1D). Overall, dinactin was selected for additional studies.

### Antibacterial properties of dinactin
Macrotetrolides, including dinactin, have been described to exhibit a wide range of biological activities, including antibacterial, antiparasitic, and anticancer[20]. The REMA assay was applied to verify its antibacterial spectrum by determining the MICs of various bacteria. It can be seen that dinactin is effective against all of the mycobacteria genera except *M. smegmatis* mc[2] 155 (MIC 32 μg/ml) (Supplementary Table 1). The dinactin also showed good activity against a panel of clinical drug-resistant isolates with a MIC of 0.5–4 μg/ml (Supplementary Table 2), indicating the potential efficacy of dinactin to combat sensitive and drug-resistant tuberculosis. Interestingly, dinactin also possessed potent antibacterial activity against *M. abscessus*, a dangerous nontuberculous mycobacteria. Moreover, dinactin exhibited significant activity against several Gram-positive strains, but showed no detectable activity against Gram-negative strains. These results agree with earlier observations and imply that dinactin might target a component widely present in Gram-positive strains, including mycobacteria. To further understand the bactericidal properties of dinactin against *M. tuberculosis* H37Rv, the MBC was determined. The MBC result indicated that dinactin had a bactericidal effect at a concentration of 1 μg/ml.

### Bactericidal action of dinactin
Dinactin exhibited an excellent inhibitory effect against *M. tuberculosis* H37Rv. Therefore, we further detected whether this ability could translate into similar bactericidal activity. The killing assay of dinactin was determined against *M. tuberculosis* H37Rv. We explored the kill kinetics of dinactin to *M. tuberculosis* H37Rv, and dinactin revealed a germicidal activity in a concentration-time dependent manner. The bactericidal activity of dinactin to *M. tuberculosis* H37Rv with a 5 log10 and 6 log10 reductions in CFU at 4 μg/ml and 8 μg/ml, respectively (Fig. 1E). Dinactin activity was higher than isoniazid activity, especially since isoniazid showed regrowth after treatment. Furthermore, in this study, a nutrient-deficient mycobacterial strain as a dormant model was also applied to evaluate dinactin activity against non-replicating mycobacteria[21]. As shown in Fig. 1F, the result revealed that dinactin was efficacious against nutrient-starved non-replicating mycobacteria. Dinactin caused approximately 4 log10 reductions of CFU under 7 days of treatment at 2 μg/ml, and the classical anti-tuberculosis drug rifampicin (1 μg/ml) was effective on non-replicating mycobacteria.

### Effect of dinactin against intracellular *M. tuberculosis* H37Rv
In this assay, the intracellular killing action of dinactin was evaluated in a THP-1-derived macrophage model. The result demonstrated that exposure to 16 × MIC of dinactin (8 μg/ml) and isoniazid (1 μg/ml) for 48 h caused at least 2 $\log_{10}$ CFU decreases in intracellular *M. tuberculosis* H37Rv (Fig. 1G). This indicates that dinactin can penetrate macrophages and suppress bacilli replication.

### The synergy of dinactin with antituberculosis drugs
The fast emergence of drug-resistant *M. tuberculosis* poses a significant challenge to TB control[22]. However, since *M. tuberculosis* can rapidly develop resistance to individual drugs, developing effective combination regimens is essential to shorten the duration of antituberculosis treatment. Therefore, checkerboard dilution assays between dinactin and multiple classes of antibiotics were performed to examine the in vitro effect of the combination, including rifampicin, isoniazid, ethambutol, linezolid, beda-quiline, PA824, and PBTZ169 (Fig. 2A–G). Interestingly, we observed that dinactin potentiated most antibiotics' activity against *M. tuberculosis* H37Rv (Fig. 2H). Dinactin displayed the best synergistic activity with rifampicin (FICI = 0.0626), accompanied by a 32-fold decrease in MIC values from 0.0156 to 0.00049 μg/ml (Supplementary Table 3). We also observed strong synergy between dinactin and isoniazid, linezolid, bedaquiline, PA824, and

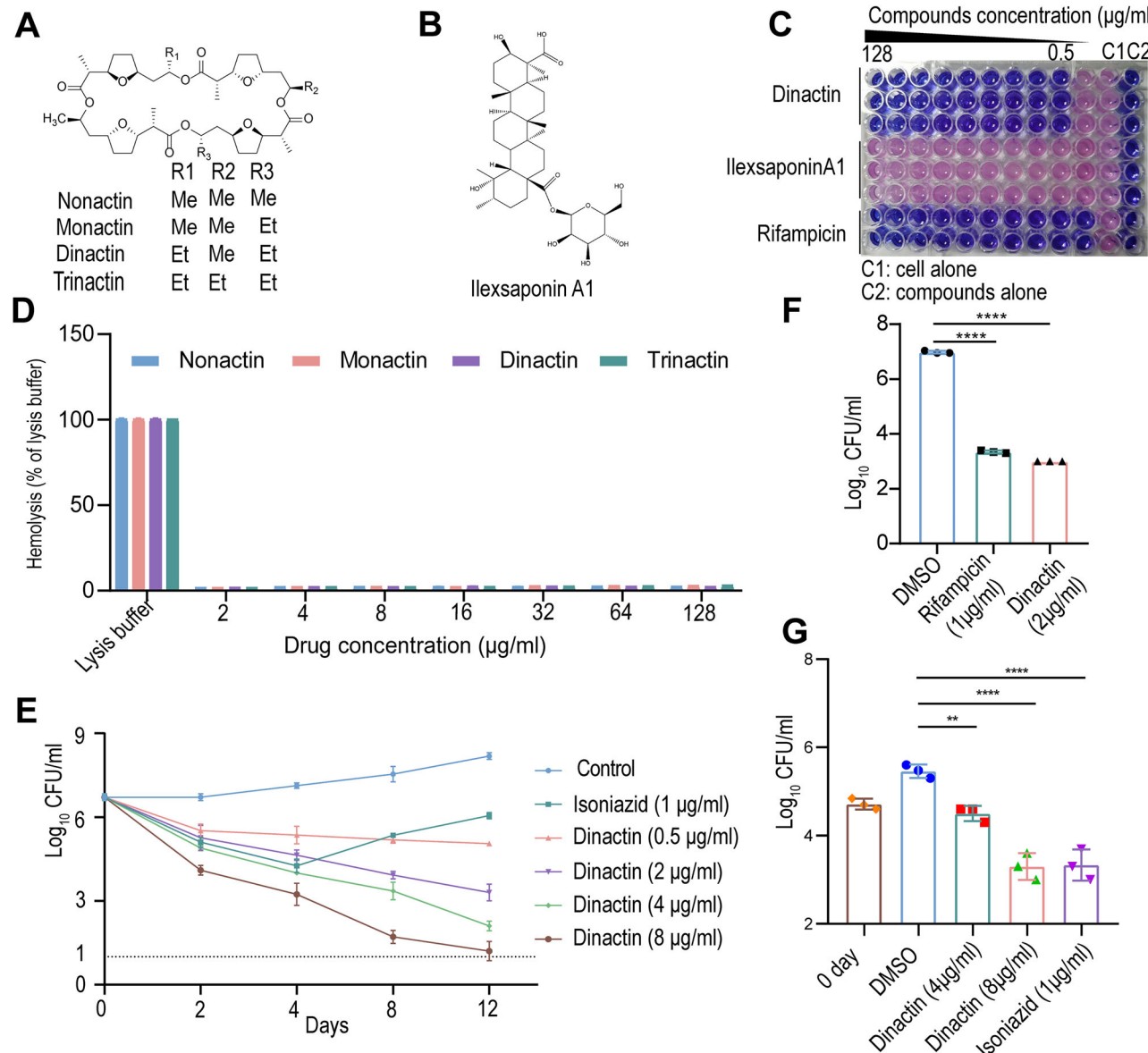

**Fig. 1 | Compound structure and antibacterial activity of dinactin. A**, **B** Structure of dinactin and its homologs and Ilexsaponin A1. **C** A representative image of resazurin reduction microplate assay was utilized for MIC determination of dinactin and ilexsaponin A1 against *M. tuberculosis* H37Rv. Rifampicin served as a positive control. **D** Hemolysis effect of four macrotetrolide compounds against ovine red blood cells. **E** Kill kinetics of various concentrations of dinactin against *M. tuberculosis* H37Rv in 7H9 medium compared with isoniazid. Dotted lines indicated the detection limit (Log10 CFU = 1). **F** Bactericidal effect of dinactin against non-replicating mycobacteria. BCG starved in a nutritionally deprived condition for 6 weeks, and then the nutrient starvation persistence model was exposed to 16 × MIC dinactin (2 μg/ml) and Rifampicin (1 μg/ml) for 7 days. **G** Intracellular bactericidal activity of dinactin on *M. tuberculosis* H37Rv. Different concentrations of dinactin and positive control isoniazid were employed for treating infected macrophages. CFUs were measured after 2 days of treatment. Experiments were performed in triplicate and carried out with at least two biological replicates. Data were analyzed using software GraphPad Prism and shown as mean ± SD, unpaired student's *t*-test or one-way ANOVA were utilized to conduct statistical significance analysis. ($P < 0.0001$****, $P < 0.001$***, $P < 0.01$**, $P < 0.05$*, $P > 0.05$, ns not significant).

PBTZ169 with FICI of 0.125 to 0.281 (Supplementary Table 3). As for ethambutol, the additive effect was found with a FICI of 0.5625.

Given the potentiation of dinactin on rifampicin and isoniazid against *M. tuberculosis* H37Rv, we further evaluated the effect by using the combination of dinactin with rifampicin or isoniazid to measure the potency of several *M. tuberculosis* drug-resistant isolates. As we expected, dinactin could reverse MDR *M. tuberculosis* isolates with the MICs being decreased from 1 to 32 μg/ml to 0.0156 to 1 μg/ml for rifampicin (Table 2) or 2–128μg/ml to 0.0313–32 μg/ml for isoniazid (Table 3). In short, combining dinactin with other antituberculosis drugs may be an optional strategy to combat *M. tuberculosis* infection. Moreover, this also implied that dinactin served as a potent antibiotic for enhancing the efficacy of antituberculosis drugs, particularly for both sensitive *M. tuberculosis* and drug-resistant mycobacteria.

## Enhanced killing of stationary-phase *M. tuberculosis* with dinactin in combination with rifampicin or isoniazid

The ability of *M. tuberculosis* to enter the stationary phase or non-replicating states, is considered the leading cause of treatment failure[23]. Considering the synergy of dinactin with rifampicin or isoniazid, we will further explore whether the sterilized effect could be enhanced by combining dinactin with rifampicin or isoniazid. Therefore, we evaluated the killing efficacy of dinactin alone or in combination with rifampicin and isoniazid at 10 × MIC against stationary-phase *M. tuberculosis* H37Rv. The result showed that dinactin was bactericidal in the stationary phase, causing more than a 3 log10 CFU decrease. This activity was comparable to that of rifampicin (4 log10 CFU killing) and better than that of isoniazid (less than 2 log10 CFU killing) (Fig. 2I). Interestingly, when dinactin was combined with

## Table 1 | Antituberculosis activity of dinactin and its homologs

| Compound | MIC (µg/ml) M. tuberculosis H37Rv | MBC (µg/ml) | IC50 Vero | Selective index |
|---|---|---|---|---|
| Dinactin | 0.5 | 1 | > 64 | > 128 |
| Nonactin | 1 | 4 | > 32 | > 32 |
| Monactin | 1 | 8 | > 64 | > 64 |
| Trinactin | 4 | 16 | > 64 | > 16 |
| Rifampicin | 0.0156 | 0.125 | > 64 | > 4096 |
| Isoniazid | 0.0625 | 1 | > 64 | > 1024 |
| Ethambutol | 1 | 4 | > 64 | > 64 |

Selective index = IC50/MIC.

isoniazid caused a rapid killing efficacy, accompanied by an approximate 3 log10 CFU drop after 20 days of treatment, compared to isoniazid alone. Moreover, a more robust killing was observed between the combination of dinactin and rifampicin, where bacterial counts were decreased by about 4 log10 CFU, compared to either dinactin or rifampicin alone.

### Efficacy of dinactin in the *Galleria mellonella* model

After elucidating the activity of dinactin in vitro assays and cell infection models, we further assessed whether a similar effect was observed in vivo. Given the excellent synergism of dinactin and rifampicin or isoniazid against wild-type and drug-resistant *M. tuberculosis* in vitro, we reasoned that the combination would be effective in vivo. Therefore, we detected the in vivo efficacy of the combination in a *Galleria mellonella* model infected with *M. tuberculosis* H37Rv (Fig. 3A). As shown in Fig. 3B, after 5 days of drug treatment, the dinactin, rifampicin, or isoniazid use alone group had 55.5, 67, or 70% survival rates, respectively. Compared with the rifampicin or isoniazid administration group, the survival rate of the larvae in the dinactin+rifampicin and dinactin+isoniazid combination group significantly increased and achieved 92 and 88%. Histopathology analysis of infected larvae from the control and drug treatment groups showed that the combination therapy group exhibited a better efficacy with less bacterial burden in larvae as indicated by acid-fast staining (Fig. 3C).The CFU count result showed that the drug therapy group exhibited a better efficacy with less bacterial burden in larvae. Compared with the PBS treatment group, after dinactin, rifampicin or isoniazid administration alone, the bacterial load in the larvae was significantly reduced by about 1.7 log10 CFU, 2.4 log10 CFU or 2.3 log10 CFU respectively (Fig. 3D). And the drug combination groups displayed better bactericidal activity in vivo than the rifampicin or isoniazid used alone, the CFU was reduced by 1.5 log10 and 1.4 log10, respectively (Fig. 3D).

### Dinactin promotes the transport of $K^+$ and $Na^+$ across the membrane

Dinactin and its homologs have been described as mobile ion carriers to produce their electrical properties on the membrane by forming positively charged complexes[24,25]. Recent research showed that the antibacterial activity of macrotetrolides (nonactin) was relative to their ability to form stable complexes with monovalent cations ($K^+$, $Na^+$). This ability enables cations across the cell membrane via passive diffusion[26]. To verify the capacity of dinactin as an ionophore, we measured the total cation contents in BCG treated with dinactin by inductively coupled plasma mass spectrometry (ICP-MS). The result indicated that treatment with dinactin significantly increased intracellular $K^+$ and $Na^+$ levels (Fig. 4A) and suggested that dinactin promotes cation transport across the membrane and shows a high selectivity for $K^+$ and $Na^+$ ions. This result is consistent with a previous study[25].

### Dinactin affects mycobacterial cell membrane permeability and fluidity

Scanning electron microscopy (SEM) was utilized to visualize bacterial membrane morphology, with wrinkled surfaces observed under the

treatment of dinactin (Fig. 4B). Furthermore, using the SYTOX Green permeability assay, we observed that dinactin induced concentration-dependent membrane permeability in BCG (Supplementary Fig. 2). Membrane-targeting compounds could insert into lipid bilayers to cause dramatic changes in membrane fluidity that disrupted the normal liquid-crystalline phase of the membrane[27]. To determine whether ionophore antibiotics affect overall membrane fluidity, we tested the fluidity of the BCG membrane under the treatment of dinactin by utilizing the membrane fluidity-sensitive dye Laurdan. The deviation of the fluorescence emission wavelength of Laurdan depends on the number of adjacent water molecules, since the penetration of the water molecule into the lipid bilayer was determined by lipid stacking density and lipid bilayer fluidity[28]. Myco-bacteria membrane fluidity can be quantified by using the Laurdan gen-eralized polarization (GP) value, ranging from $-1$ (most fluid and disordered) to $+1$ (most rigid and ordered)[28]. The addition of dinactin induced a concentration-dependent reduction in Laurdan GP in BCG. The result indicated an increase in membrane fluidity (Fig. 4C), similar to the positive control, benzyl alcohol (BA), a known membrane fluidizer. Inter-estingly, increased membrane permeability and destabilization will affect bacilli surface potential[29].

### Dinactin disrupts the proton motive force

*M. tuberculosis*, an obligate aerobe, depends on its ETC for biological energy production and the *M. tuberculosis* ETC, as a potential target, has sparked the interest of some researchers[30]. The antibacterial activity of ionophore antibiotics was connected with its characteristics that mediated ions accu-mulation or efflux in intracellular, causing the disruption of membrane potential and membrane conductance[24,26]. We, therefore, investigated if the membrane potential ($\Delta\psi$) would be dissipated after dinactin addition to whole bacteria, as seen by the classic uncoupler. The fluorescent probe 3,30-diethyloxacarbocyanine iodide (DiOC2(3)) was used to determine the membrane potential of BCG treated with dinactin. DiOC2(3) was a fluor-escent dye without membrane permeability, and the fluorescence shifted from green to red with increased membrane potential. As shown in Fig. 4D, dinactin, in a dose-dependent manner, induced the collapse of membrane potential compared with the DMSO control after being treated for 2 h. Meanwhile, as expected, the addition of the classical uncoupler CCCP did dissipate the membrane potential. The proton motive force (PMF) is essential for maintaining the viability of wild and non-replicating *M. tuberculosis*[31]. PMF is an electrochemical gradient at the cytoplasmic membrane that consists of both $\Delta\psi$ and $\Delta pH$ (proton gradient)[32]. Subse-quently, we also determined the membrane proton conductance by evalu-ating $\Delta pH$, a major component of the PMF in BCG, using the fluorescent dye BCECF-AM, a pH-sensitive probe, and the fluorescence intensity is proportional to intracellular pH[33]. BCECF-AM was nonfluorescent. Once within the cell, nonspecific esterases hydrolyze the nonfluorescent AM ester precursor, yielding the fluorescent BCECF, pH-sensitive indicator. We observed a significant fluorescence decrease when different concentrations of dinactin were added to cultures (Fig. 4E), signifying the dissipation of the $\Delta pH$. After adding the protonophore CCCP also detected a similar effect, this data suggests that dinactin can disrupt the PMF.

### Dinactin interferes with ATP homeostasis and does not inhibit oxygen consumption

The ATP homeostasis of *M. tuberculosis* depends on the PMF[31]. To deter-mine whether dinactin might disrupt the energy metabolism homeostasis of mycobacteria, we measured the early intracellular ATP levels under the drug treatment. Dinactin-treated BCG showed a reduction in intracellular ATP levels as early as 24 h after treatment, and dinactin triggered ATP plummet in a dose-dependent manner (Fig. 4F). Previously, it has been reported that non-replicating mycobacteria were viable with only a tenth of ATP levels compared to replicating bacilli, but further ATP exhaustion will result in efficient killing[34]. To verify this effect, we further determined the ATP concentration of non-replicating mycobacteria as mentioned above, and a similar decrease was also observed (Supplementary Fig. 3). In particular, this

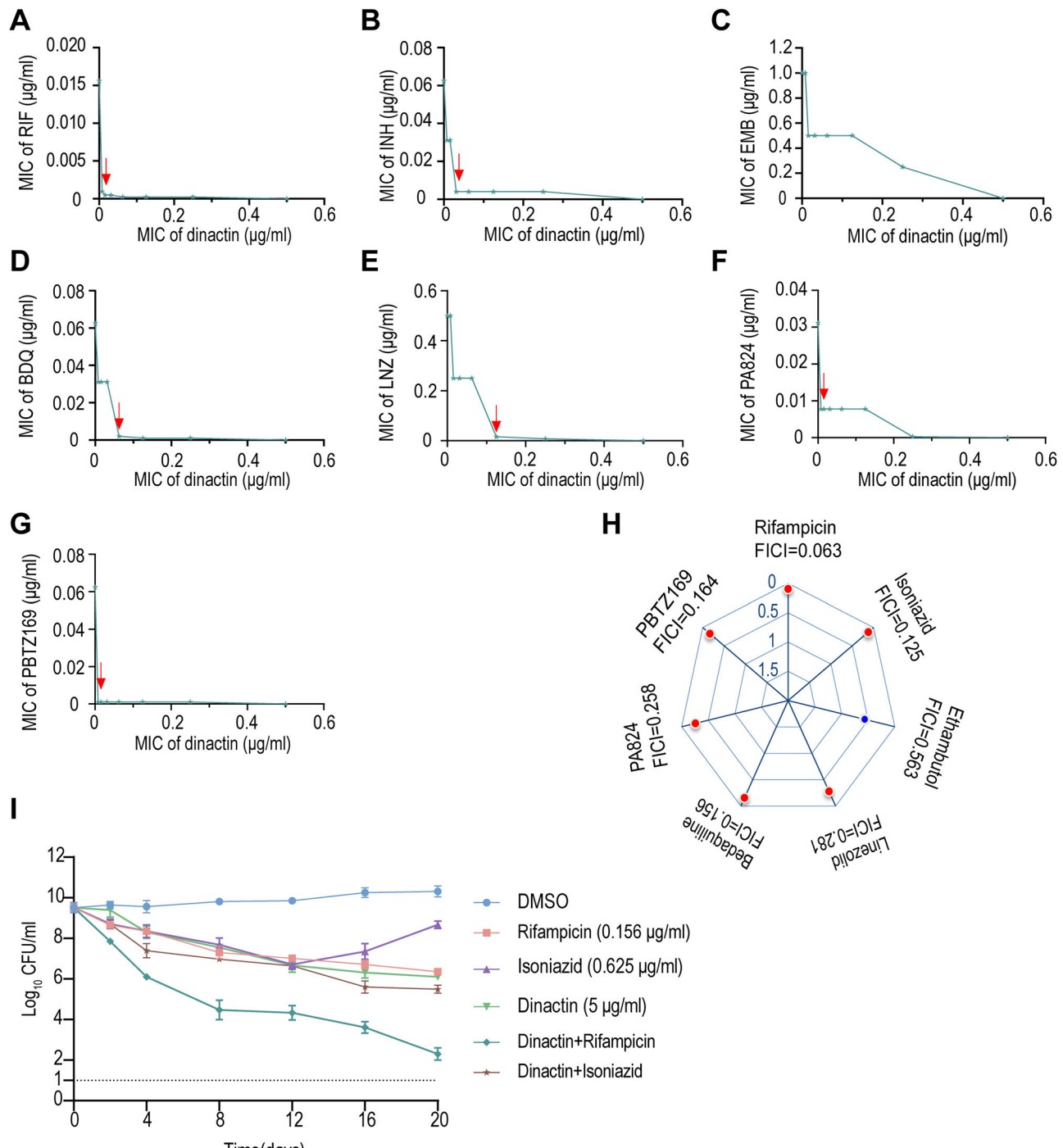

**Fig. 2 | Pharmacodynamic interactions between dinactin and different anti-tuberculosis drugs against drug-susceptible and -resistant *M. tuberculosis.*** **A–G** Checkerboard assay revealed that the synergism of the combination between dinactin with RIF (rifampicin), INH (isoniazid), BDQ (bedaquiline), LNZ (line-zolid), PA824, and PBTZ169, except EMB (ethambutol). The red arrow pointed to the drug concentration with the best synergy. **H** Summary of the synergistic effects of dinactin with these antituberculous drugs and data were represented by FICI. FICI ≤ 0.5 shown as synergism. **I** Kill kinetic of dinactin (5 μg/ml, 10 × MIC) with rifampicin (0.156 μg/ml, 10 × MIC) and isoniazid (0.625 μg/ml, 10 × MIC) alone or in combination with rifampicin and isoniazid against stationary-phase *M. tuber-culosis*, respectively. Dotted lines indicate the detection limit (Log₁₀ CFU = 1). Experiments were performed in triplicate and carried out with at least two biological replicates and data were shown as mean ± SD.

decrease in intracellular ATP content was correlated with its bactericidal activity. It is all known that *M. tuberculosis* is an obligate aerobic pathogen, that could produce ATP mainly through the oxidative phosphorylation (OxPhos) pathway[35]. Therefore, inhibition of the respiration of myco-bacteria will interfere with energy metabolism and growth. Thus, we investigated whether dinactin inhibited BCG respiration by monitoring oxygen consumption. Methylene blue was utilized as a qualitative indicator of dissolved oxygen level, and as visualized for BCG (Fig. 4G), the dissolved oxygen level was unaffected under the treatment of dinactin at a high concentration after 120 h. This data reveals that the reduction in ATP levels is not caused by inhibiting dinactin against mycobacterial respiration.

**Table 2 | The synergistic effect of dinactin with rifampicin against drug-resistant *M. tuberculosis***

| Strains | Drug | MIC (µg/ml) | | FIC | FICI | Remarks |
|---|---|---|---|---|---|---|
| | Combination | Alone | Combination | | | |
| CR-1 | Dinactin | 0.5 | 0.0625 | 0.125 | 0.156 | synergy |
| | Rifampicin | 8 | 0.25 | 0.03125 | | |
| CR-2 | Dinactin | 1 | 0.25 | 0.25 | 0.5 | synergy |
| | Rifampicin | 1 | 0.25 | 0.25 | | |
| CR-5 | Dinactin | 1 | 0.125 | 0.125 | 0.156 | synergy |
| | Rifampicin | 2 | 0.0625 | 0.03125 | | |
| CR-6 | Dinactin | 0.5 | 0.0625 | 0.125 | 0.375 | synergy |
| | Rifampicin | 2 | 0.5 | 0.25 | | |
| CR-7 | Dinactin | 1 | 0.25 | 0.25 | 0.313 | synergy |
| | Rifampicin | 2 | 0.125 | 0.0625 | | |
| CR-8 | Dinactin | 4 | 0.25 | 0.0625 | 0.0781 | synergy |
| | Rifampicin | 4 | 0.0625 | 0.0156 | | |
| CR-11 | Dinactin | 4 | 0.25 | 0.0625 | 0.0938 | synergy |
| | Rifampicin | 32 | 1 | 0.0313 | | |
| CR-12 | Dinactin | 4 | 0.125 | 0.0313 | 0.0938 | synergy |
| | Rifampicin | 2 | 0.125 | 0.0625 | | |
| CR-14 | Dinactin | 2 | 0.125 | 0.0625 | 0.125 | synergy |
| | Rifampicin | 2 | 0.125 | 0.0625 | | |
| CR-15 | Dinactin | 2 | 0.03125 | 0.0156 | 0.0781 | synergy |
| | Rifampicin | 2 | 0.125 | 0.0625 | | |
| CR-18 | Dinactin | 4 | 0.25 | 0.0625 | 0.188 | synergy |
| | Rifampicin | 16 | 2 | 0.125 | | |
| CR-19 | Dinactin | 2 | 0.0625 | 0.0313 | 0.0938 | synergy |
| | Rifampicin | 2 | 0.125 | 0.0625 | | |
| CR-21 | Dinactin | 4 | 0.25 | 0.0625 | 0.0713 | synergy |
| | Rifampicin | 2 | 0.0156 | 0.0078 | | |
| CR-22 | Dinactin | 4 | 0.5 | 0.125 | 0.188 | synergy |
| | Rifampicin | 1 | 0.0625 | 0.0625 | | |

## Dinactin alters the NADH/NAD⁺ redox balance of mycobacteria and induces the generation of ROS

The electron transport chain (ETC) and ATP homeostasis were essential to both actively growing cultures and persistent bacilli[36–38]. According to the above result, dinactin disrupted the PMF of mycobacteria, and the ETC was essential for *M. tuberculosis* to maintain the PMF. We, therefore, speculated that dinactin might also affect the function of the respiratory chain. The reduction assay of resazurin was executed to verify the speculation[39]. Indeed, dinactin diminished the reductive activity of BCG and indicated inhibition of ETC (Fig. 4H). However, this inhibition did not affect the respiration of BCG (Fig. 4G). The adverse effects of dinactin on cytoplasmic membranes might also cause inefficient turnover in the electron transfer from NADH to $NAD^+$. To further assess our conjecture, we treated BCG with dinactin, and thioridazine was used as a positive control, detecting the ratio of NADH/$NAD^+$. The result shows that dinactin at 0.25 or 1 µg/ml significantly elevated the ratio of NADH/$NAD^+$ (Fig. 4I), indicating that dinactin disrupted the intracellular NADH/$NAD^+$ redox balance, driving the BCG into a more reductive state and accumulation of more reduction equivalents was lethal to bacteria. Furthermore, the drug disrupting the ETC would cause a non-enzymatic transfer of electrons to oxygen to produce superoxide[30]. Bacter-icidal antibiotics can trigger the production of deleterious hydroxyl radicals in pathogens, and the damage to cytomembrane homeostasis will also result in the accumulation of reactive oxygen species (ROS)[40,41]. We thus measured the ROS level of treated BCG using a ROS-sensitive fluorescent probe. As a bactericidal drug, dinactin stimulated the accumulation of intracellular ROS

compared with the control group (Fig. 4J). Collectively, our data showed that natural product dinactin targeted the mycobacteria membrane, enhancing membrane fluidity and permeability, causing cell membrane damage, disrupting the redox balance, and collapsing the intracellular homeostatic and physiological functions (Fig. 4K).

## Identification of the dinactin target

In order to gain insight into the mechanism of action of dinactin and identify the molecular target, spontaneously resistant strains against dinactin were selected to pinpoint the exact mutations and search for the anticipated target that might be contained in these mutant genes. Five BCG mutants were isolated and 4–8-fold resistant to dinactin. In addition, these mutants remained unchanged relative to other antituberculosis antibiotics, including rifampicin, isoniazid, and ethambutol (Table 4). Both these mutants and the parental strain were subjected to whole-genome sequencing to further investigate the single-nucleotide polymorphisms (SNPs) responsible for dinactin resistance.

Comparing the mutants and parental BCG, four mutants had the same SNP mutation in *cpsA*, resulting in amino acid substitution Asp347Gly (Table 4). Moreover, the nucleic acid and amino acid sequences of *cpsA* and *CpsA1* in BCG shared an equivalent amino acid sequence with *Rv3484* and *Rv3267* in *M. tuberculosis* H37Rv. *cpsA* protein belongs to the widespread LytR-Cps2A-Psr (LCP) family protein, which also contains another protein *Rv3267* (here named *CpsA1*). The LCP protein is responsible for the func-tion of the covalent attachment of the two major components of the

**Table 3 | The synergistic effect of dinactin with isoniazid against drug-resistant *M. tuberculosis***

| Strains | Drug | MIC (µg/ml) | | FIC | FICI | Remarks |
|---|---|---|---|---|---|---|
| | Combination | Alone | Combination | | | |
| CR-1 | Dinactin | 0.5 | 0.0625 | 0.125 | 0.25 | synergy |
| | Isoniazid | 8 | 1 | 0.125 | | |
| CR-2 | Dinactin | 1 | 0.03125 | 0.03125 | 0.281 | synergy |
| | Isoniazid | 4 | 1 | 0.25 | | |
| CR-5 | Dinactin | 1 | 0.25 | 0.25 | 0.325 | synergy |
| | Isoniazid | 2 | 0.25 | 0.125 | | |
| CR-6 | Dinactin | 0.5 | 0.125 | 0.25 | 0.313 | synergy |
| | Isoniazid | 4 | 0.25 | 0.0625 | | |
| CR-7 | Dinactin | 1 | 0.125 | 0.125 | 0.188 | synergy |
| | Isoniazid | 8 | 0.5 | 0.0625 | | |
| CR-8 | Dinactin | 4 | 0.5 | 0.125 | 0.156 | synergy |
| | Isoniazid | 2 | 0.0625 | 0.0313 | | |
| CR-11 | Dinactin | 4 | 1 | 0.25 | 0.313 | synergy |
| | Isoniazid | 8 | 0.5 | 0.0625 | | |
| CR-12 | Dinactin | 4 | 0.25 | 0.0625 | 0.125 | synergy |
| | Isoniazid | 8 | 0.5 | 0.0625 | | |
| CR-14 | Dinactin | 2 | 0.25 | 0.125 | 0.141 | synergy |
| | Isoniazid | 2 | 0.03125 | 0.0156 | | |
| CR-15 | Dinactin | 2 | 0.125 | 0.0625 | 0.185 | synergy |
| | Isoniazid | 2 | 0.25 | 0.125 | | |
| CR-18 | Dinactin | 4 | 0.25 | 0.0625 | 0.125 | synergy |
| | Isoniazid | 2 | 0.125 | 0.0625 | | |
| CR-19 | Dinactin | 2 | 0.5 | 0.25 | 0.5 | synergy |
| | Isoniazid | 128 | 32 | 0.25 | | |
| CR-21 | Dinactin | 4 | 0.5 | 0.125 | 0.156 | synergy |
| | Isoniazid | 128 | 4 | 0.0313 | | |
| CR-22 | Dinactin | 4 | 0.5 | 0.125 | 0.141 | synergy |
| | Isoniazid | 4 | 0.0625 | 0.0156 | | |

mycobacterial cell wall, arabinogalactan (AG) and peptidoglycan (PG)[42]. In addition, the combined deficiency of *CpsA1* and *cpsA* could be synthetic lethality and this suggests that the *CpsA1* and *cpsA* genes appeared to exhibit partially overlapping essential activities in *M. tuberculosis*[42]. To explore the mechanism of action of dinactin relative to dinactin resistance, we overexpressed the *cpsA* in wild-type BCG and detected the susceptibility of these overexpressing strains to dinactin. Compared to the wild-type strain, overexpression of wild-type *cpsA* (pMV261-*cpsA*) caused a twofold higher MIC for dinactin (Fig. 5A), and the overexpressed strain could grow when plated on 7H11 agar containing 2 × MIC of dinactin (Fig. 5B). Furthermore, we also overexpressed *CpsA1* in BCG, resulting in a fourfold increase in MIC to dinactin (Fig. 5C). This result was verified on a resistant plate containing 4 × MIC of dinactin (Fig. 5D).

**Genetic validation of *cpsA* as a target of dinactin**
We constructed the *cpsA*-deletion mutant in BCG and complemented the strain to evaluate the resistance to dinactin. The knock-out mutant resulted in high resistance levels when plated onto 7H11 plates containing dinactin, whereas the complemented strain could not grow on the resistant plates (Fig. 5E). We also made efforts to construct a *CpsA1*-deletion mutant in BCG. However, we failed to obtain a mutant strain. To our knowledge, dinactin is the first inhibitor that targets the LCP family protein in *M. tuberculosis*. Since the LCP family protein is widespread in Gram-positive bacteria, this may be answerable for the selective activity of dinactin against Gram-positive bacteria. However, our findings are not definitive, and we

only tentatively identified the LCP protein of *M. tuberculosis* as the target of dinactin. Hence, further detailed biochemical experiments are needed to validate our conclusion.

**Determination of the interaction between dinactin and cpsAΔTM**
Despite expressing a large amount of pure, soluble cpsAΔTM, our attempts to obtain the experimental crystal structure of cpsA have proven unsuccessful. Therefore, we generated a cpsA homology model using the online AlphaFold server (Fig. 6A)[43]. The docking result showed that dinactin was observed to occupy the active site of the protein of cpsA (Fig. 6B). Subsequently, we found that the receptor-ligand interaction between dinactin and cpsA was determined by the Asp347, Pro348, Pro136, Thr138, Ser192, Arg352, Thr349, Asp442, Gly440, Ile441, Thr420, Asp461, Asn463, Phe464, Ser465, Ile346, and His145 residues in the binding pocket (Fig. 6C). The Thr138 and Asn463 residues might have an important effect on this interaction by forming a hydrogen bond with dinactin. In addition, the hydrophobic interactions between dinactin and the Pro136, Pro348, and Ile441 residues are also of interest because hydrophobic interactions play an important role in maintaining protein folding.

To further investigate the biological activity of dinactin with target, we performed the MST assay to measure their binding interaction. The $K_d$ value reflects the affinity between ligand and protein, and the result revealed that dinactin could bind cpsAΔTM in a dose-dependent manner with a $K_d$ value of 0.13 ± 0.07 µM (Fig. 6D). Furthermore, we detected the affinity of dinactin with cpsAΔTM$^{Asp347Gly}$, a weak interaction was observed and

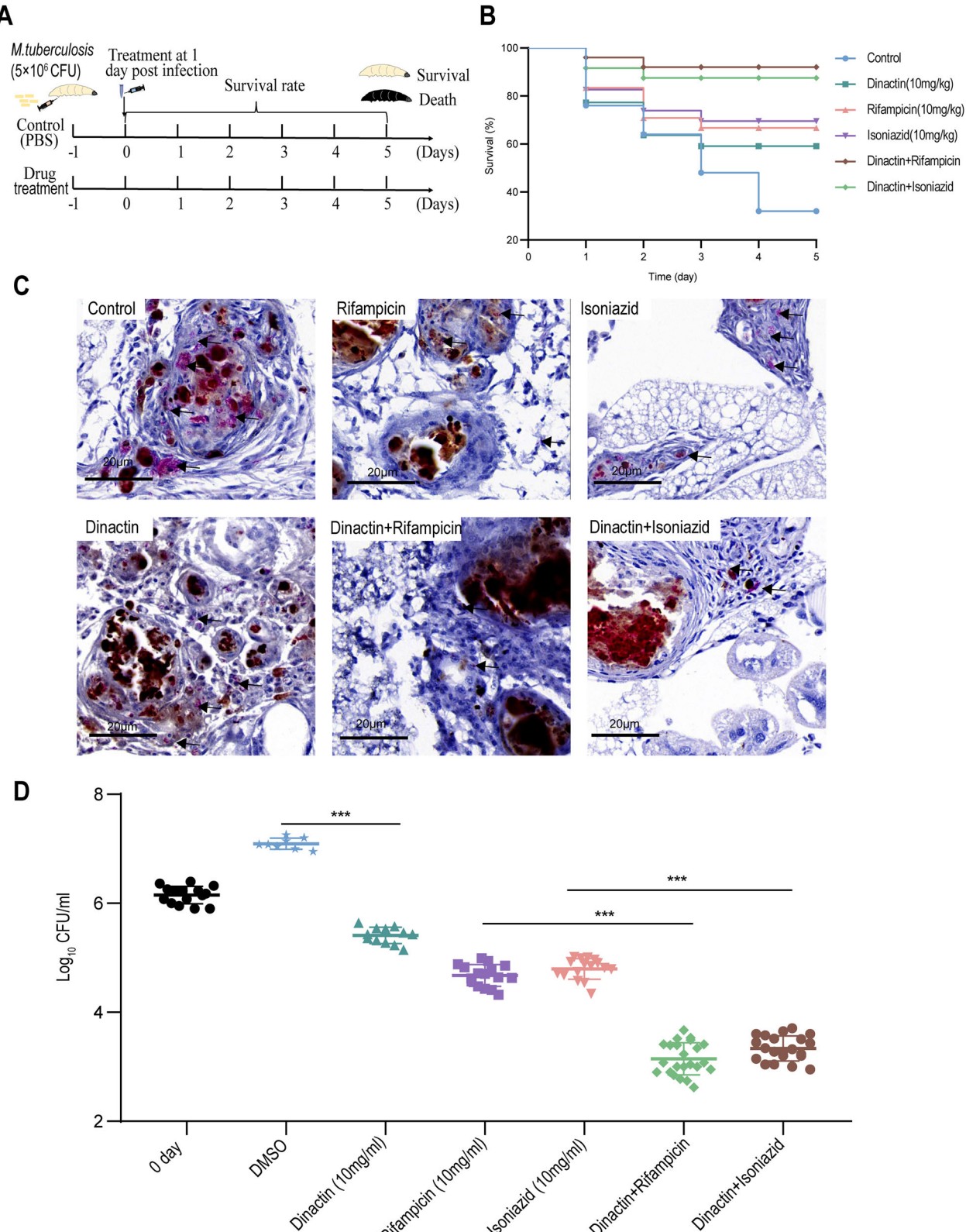

**Fig. 3 | The therapeutic efficacy of dinactin in the *Galleria mellonella* model infected with *M. tuberculosis* H37Rv. A** Scheme of the experimental protocols for the *Galleria mellonella* infection model. **B** Infected *Galleria mellonella* larvae were treated with dinactin, rifampicin, isoniazid alone or the combination of dinactin with rifampicin or isoniazid at 24 h post-infection, and survival was monitored for 5 days. **C** Histopathological analysis of infected Galleria mellonella larvae after drug treatment. Survival larvae were stained with Ziehl–Neelsen dye for acid-fast *M. tuberculosis*. Bacilli were stained pink and black arrow points to stained bacilli. **D** Bacterial load in the infected larvae with different drug treatments. Data were shown as mean ± SD and the unpaired Student's *t*-test or one-way ANOVA was used to conduct statistical significance analysis. ($P < 0.001$***).

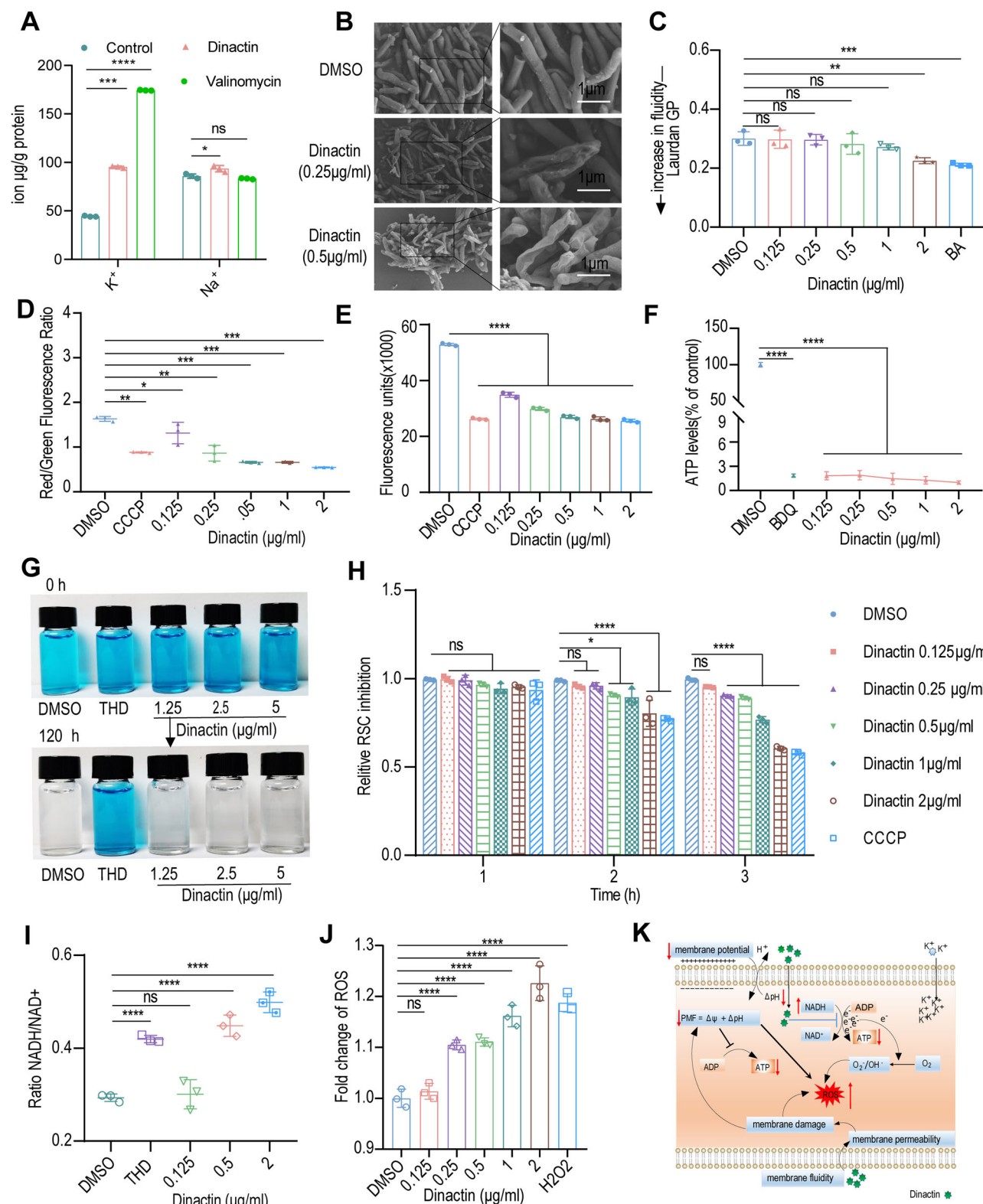

obtained a high $K$d value of $502.2 \pm 34\,\mu M$ (Fig. 6E). Interestingly, the mutation of the amino acid resulted in a rapid increase in $K$d value. This indicated that the amino acid Asp347 might play an important role in their interaction. Due to the partially functional compensation of CpsA1 and cpsA, we also tested whether there is an interaction between CpsA1ΔTM and dinactin, and the data showed that dinactin could interact with CpsA1ΔTM with a $K$d value of $7.82 \pm 1.8\,\mu M$ (Supplementary Fig. 4). The

results suggest that there was an interaction between dinactin and the two LCP family proteins and show preferential selectivity for the cpsA protein.

## Discussion

Continued efforts have been taken to combat MDR/XDR-tuberculosis and improve the chemotherapy of tuberculosis, including developing innovative therapies and novel antituberculosis drugs, which are required to control

**Fig. 4 | Dinactin disrupted the proton motive force and energy metabolism of mycobacteria. A** ICP-MS analysis of the levels of $Na^+$ and $K^+$ in BCG cells after dinactin treatment (0.25 µg/ml, 2×MIC) for 24 h. **B** Mycobacteria membrane morphology under the treatment of the indicated concentrations of dinactin for 24 h, visualized with SEM. **C** Mycobacteria membrane fluidity treated with dinactin for 3 h was evaluated based on Laurdan generalized polarization (Laurdan GP). BA benzyl alcohol. **D**, **E** Dinactin disrupted the proton motive force of mycobacteria. The change of Δψ and ΔpH were detected by using different concentrations of dinactin treated for 24 h in BCG. CCCP served as a positive control. Reduced-Red/Green fluorescence ratio presented the decrease of membrane potential, and the reduction of the fluorescence intensity of BCECF-AM indicated the dissipation of the transmembrane proton gradient. **F** Decreased level of intracellular ATP in BCG treated with dinactin for 24 h. Bedaquiline (BDQ) served as a positive control.

**G** Dinactin did not inhibit mycobacteria respiration. THD: thioridazine. **H** Determination of the activity of dinactin against the mycobacteria electron transport chain by reduction of resazurin to resofurin. CCCP uncoupled the respiratory chain from the proton gradient and was utilized as positive. **I** The change of redox homeostasis of BCG were detected by using diffferent concentrations of dinactin treated for 1, 2, and 3 h, respectively. THD thioridazine. **J** The level of intracellular ROS were detected by using different concentrations of dinactin treated for 24 h in BCG. H2O2 was the positive control. **K** Mechanism of action of dinactin in mycobacteria. Experiments were performed in triplicate and carried out with at least two biological replicates. Data were shown as mean ± SD and unpaired student's $t$-test or one-way ANOVA was used to conduct statistical significance analysis. ($P < 0.0001$****, $P < 0.001$***, $P < 0.01$**, $P < 0.05$*, $P > 0.05$, ns not significant).

### Table 4 | Identification of dinactin resistance mutations

| | SNP | Base | Amino acid | | MIC (µg/ml) | | | |
|---|---|---|---|---|---|---|---|---|
| Strain | Location | Change | Substitution | Annotation | Dinactin | RIF | INH | EMB |
| BCG | nd | nd | nd | nd | 0.125 | 0.00098 | 0.0625 | 2.0 |
| DR-mut1 | 2192501 | A161C | Asp54Ala | Uncharacterized protein | 1.0 | | | |
| | 2203278 | C166G | Arg56Gly | Putative protein | | | | |
| DR-mut2 | 3881452 | A1040G | Asp347Gly | LytR family protein | 1.0 | 0.00098 | 0.0625 | 2.0 |
| | 2203278 | C166G | Arg56Gly | Putative protein | | | | |
| DR-mut3 | 3881452 | A1040G | Asp347Gly | LytR family protein | 0.5 | 0.00098 | 0.0625 | 2.0 |
| DR-mut4 | 3881452 | A1040G | Asp347Gly | LytR family protein | 0.5 | 0.00098 | 0.0625 | 2.0 |
| | 2192501 | A161C | Asp54Ala | Uncharacterized protein | | | | |
| DR-mut5 | 3881452 | A1040G | Asp347Gly | LytR family protein | 0.5 | 0.00098 | 0.0625 | 2.0 |

*RIF* rifampicin, *INH* isoniazid, *EMB* ethambutol, *nd* no data.

this widespread epidemic. Bacterial-derived antibiotics play an essential action in the antituberculosis area[44]. Numerous bacteria produce a variety of natural products with antimicrobial activity against competing microbes, acting like small factories and producing many kinds of highly effective and structurally diverse natural compounds. Since bacteria with the same ecological niche tend to be affected by some broad-spectrum antibiotics and express resistance, this phenomenon will result in many potential compounds that may have been overlooked[45]. In this study, through a series of experiments, one hit compound with whole-cell activity against *M. tuberculosis* was finally screened from a natural products library and identified a target of dinactin, one of the macrotetrolides. Macrotetrolide antibiotics were first reported as an ionophore. The bactericidal effect of dinactin may be closely related to the electrical properties of dinactin involving mycobacterial ETC destabilization and inhibition of energy metabolism. Dinactin also interacted with the cytomembrane of mycobacteria and disrupted membrane function. Given the cationic affinity and lipophilicity of dinactin, this character may be responsible for the membrane-damaging effect of dinactin. Moreover, this damage in the mycobacteria plasma membrane will disrupt the PMF and reduce the ATP level. The efflux pumps transport activity of drugs is closely related to the phenotypic drug tolerance of *M. tuberculosis*[46]. Moreover, a critical point of efflux pumps is its activity mainly depends on PMF or ATP, which intrinsically links drug efflux to ETC or energy metabolism of *M. tuberculosis*[47,48]. The inhibitory effect of dinactin on ETC may reduce efflux and increase intracellular drug concentrations. This effect may be related to the bactericidal effect of dinactin against drug-resistant *M. tuberculosis* and the excellent synergy shown by dinactin with rifampicin and isoniazid against MDR strains. Since most reactions involving the generation of electrochemical gradients are redox-driven, such as the energetic precursors NADH, which is oxidized to release energetic electrons, this mainly depends on the reduction potential of the particular redox pair[38]. The data revealed that dinactin could elevate the ratio of NADH/NAD$^+$, indicating that the intracellular redox balance is shifted to a more reducing state. Raising the NADH/NAD$^+$ ratio disequilibrated intracellular redox homeostasis and might also concomitantly inhibit

NADH metabolic pathways[49]. The change might be due to the inhibition of ATP generation by dinactin, which would exert back pressure on the mycobacterial respiratory chain to a certain extent, thereby impeding the respiratory electron flow and diminishing rates of NADH oxidation. Furthermore, normal respiration throughout the bacillus suggests a relatively stable ETC flux. This suggests that dinactin does not completely inhibit the electron transport of *M. tuberculosis*. Mycobacterial respiration was driven by PMF and yielded adequate ATP to maintain the viability of dormant bacilli, which also depended on redox equilibrium for survival[50]. The PMF is an important link that intrinsically connects the components of the *M. tuberculosis* respiratory system. Thus, developing innovative drugs targeting the respiratory chain of *M. tuberculosis* is a promising strategy.

The phenotypic experiments showed that dinactin exhibited antituberculosis activities against both replicating and non-replicating mycobacteria and clinical isolates of drug-resistant strains. This indicated that dinactin might target a new and conserved pathway in *M. tuberculosis*. Thus, further experimental verification of dinactin against *M. tuberculosis* in different models is necessary. When bacilli enter the host, they are usually endocytosed and internalized by alveolar macrophages. These germs could evade host immune clearance and persist in the lungs[51]. The inhibitory activity of dinactin against intracellular *M. tuberculosis* was executed in a THP-1 macrophage infection model. Successful treatment regimens for TB rely on drug combinations, and an ideal new antituberculosis drug should display synergy with approved drugs[2]. Using a subinhibitory concentration of dinactin could enhance the efficacy of several antituberculosis drugs, including two frontline antibiotics, rifampicin and isoniazid. We also evaluated its antituberculosis activity in a *G. mellonella* model. However, it is essential to further evaluate the efficacy of dinactin in an acute and chronic mouse infection model or another animal model. Postgenomic technologies facilitate the discovery of novel antituberculosis compounds with new targets and metabolic pathways in *M. tuberculosis*[11]. The results of whole-genome sequencing of several spontaneously resistant strains of dinactin indicated that four out of five mutants had a mutation in *cpsA* (Asp347Gly). The mutants showed sensitivity to frontline antituberculosis drugs that was

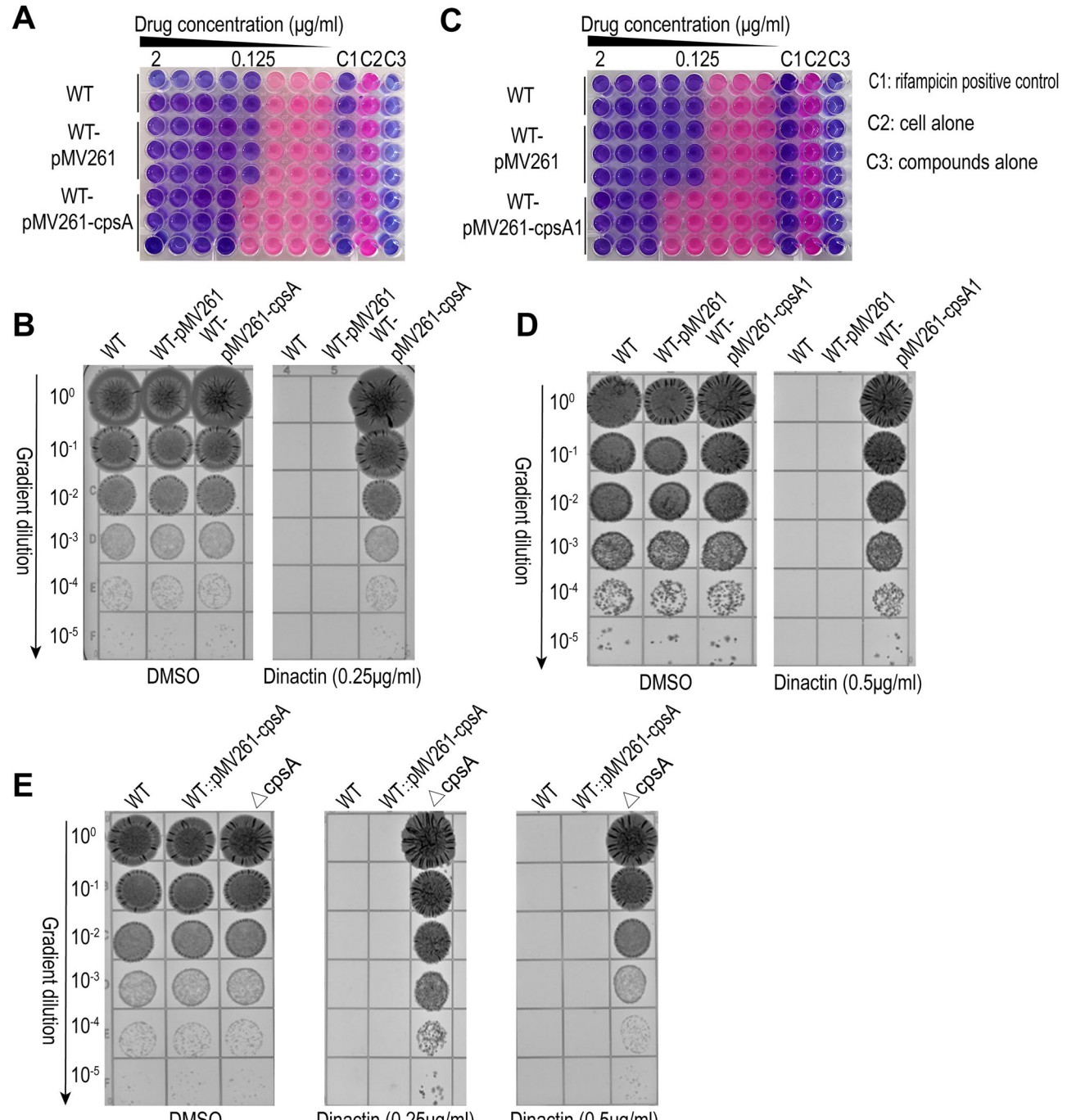

**Fig. 5 | cpsA gene was identified as a target of dinactin. A, C** Determination of the MIC of dinactin against the overexpression of the members of the LCP family protein by using resazurin reduction microplate assay. **B, D** Exponentially growing BCG or BCG carrying cpsA (**B**) or CpsA1 (**C**) gene on plasmid pMV261 were serially diluted and plated on 7H11 plates containing the appropriate concentration of dinactin. **E** The cpsA knock-out mutant could grow on the dinactin-resistant 7H11 plates.

closely relative to the action of dinactin against drug-resistant *M. tuberculosis*. Furthermore, the consistent missense mutation in the dinactin spontaneously resistant of mycobacteria implies that dinactin might target the *cpsA* protein. Subsequently, further verification was carried out through overexpression and knock-out assay. However, the knock-out mutant still exhibited susceptibility to high concentrations of dinactin, which revealed that dinactin might have other targets or pathways to kill mycobacteria.

The unique cell wall of mycobacteria is essential for its viability and is the target of many clinically used antituberculosis drugs and inhibitors under development. The main components of the mycobacterial cell wall

are the PG (peptidoglycan) layer, mycolic acid (MA) and arabinogalactan (AG). The main structural elements of the cell wall consist of a cross-linked network of PG in which some of the muramic acid residues are covalently attached to a complex polysaccharide, AG. However, how did the attachment of AG to PG, remains fragmented at best. It remains to be clearly identified the key ligase responsible for the cross-linking of AG and PG in the *M. tuberculosis* cell wall. Recently, several studies have demonstrated that the LCP family protein is responsible for the ligation of AG or wall teichoic acids or capsular polysaccharides to PG and LCP family proteins are widespread in Gram-positive bacteria, in contrast, they are absent in most

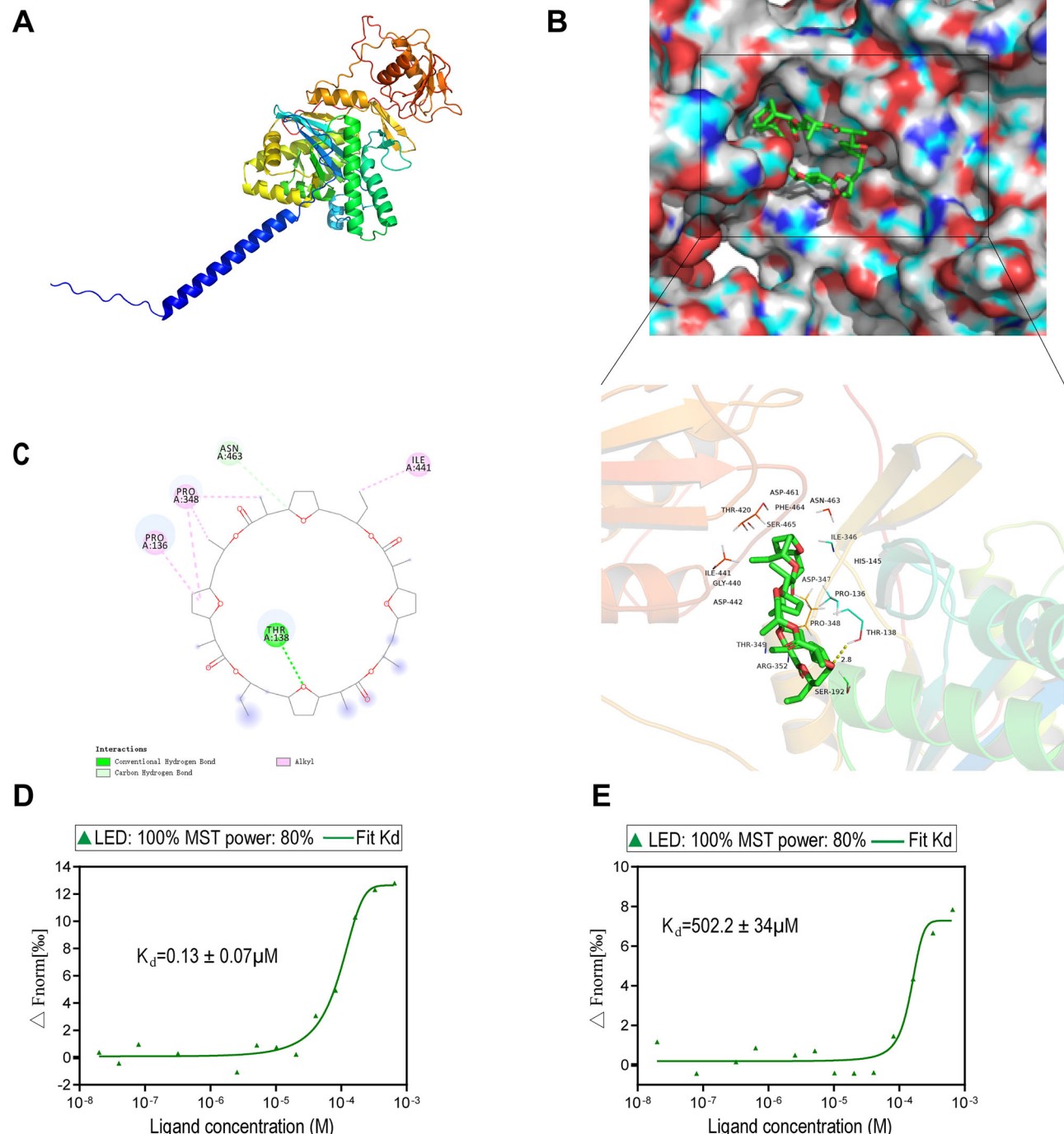

**Fig. 6 | Molecular docking and binding affinity analysis of the interaction of dinactin and cpsA. A** The predicted structure of cpsA. **B, C** The lowest-energy interaction pattern represents the docking complexes between dinactin and the cpsA protein model. Predicted hydrogen bonds are shown in green, light cyan represents predicted carbon-hydrogen bond and pink represented the hydrophobic interaction of dinactin with the amino acids in the binding pocket of cpsA protein. **D, E** Binding affinity curves detected by microscale thermophoresis for monitoring the interaction between cpsAΔTM or cpsAΔTM [Asp347Gly] and dinactin. Dinactin interacted with cpsAΔTM in a dose-response manner with KD of 0.13 ± 0.07 μM (**D**). While dinactin bound to the point mutant protein cpsAΔTM [Asp347Gly], there was a significant decrease in affinity (**E**). The MST assay were performed in triplicate, and each data represented the mean of three replicates.

Gram-negative bacteria[42,52,53]. Amongst the mycobacteria tested here, the *cpsA* protein shares a high amino acid sequence identity except *M. smegmatics* mc[2]155, which shares only 28% identity (Supplementary Fig. 5). This may explain why the tested mycobacteria were susceptible to dinactin, while dinactin displays poor inhibition to *M. smegmatics* mc[2]155. Additionally, other mycobacteria, such as *M. abscessus*, share 60% identity with the *M. tuberculosis* ortholog and is susceptible to dinactin (Supplementary Table 1 and Supplementary Fig. 5). The two major LCP family proteins in *M.*

*tuberculosis*, CpsA1 and cpsA, have been reported to be responsible for ligating AG to PG[42]. Although researchers further revealed that CpsA1 might be the essential enzyme, at the meantime, cpsA was thought to have a minor role in *M. tuberculosis*. Genetic assay in *M. tuberculosis* showed that the combined deletion of *CpsA1* and *cpsA* was synergistic lethality, this suggested that the two genes possessed partially overlapping essential activities in *M. tuberculosis* rather than redundant functions[42,54]. Further verification of the biochemical experiment revealed that a slight decrease in

AG attachment to PG in the *cpsA* deleted mutant background[55]. Although the *cpsA* knock-out did not affect mycobacteria growth, the mutant failed to establish infection in the mouse model compared with the *M. tuberculosis* H37Rv, and this defect could not be compensated by *CpsA1*[56]. Moreover, previous studies reported that the essential role of *cpsA* in *M. tuberculosis* to evade the elimination of host innate immunity by inhibiting NADPH oxidase and LAP[55]. These data suggested that *cpsA* plays a key role in mycobacterial infection and persistence within host tissues. Thus, developing innovative antituberculosis drugs targeting LCP proteins may be an ideal pathway, since these proteins are closely related to the cell wall synthesis and in vivo virulence of *M. tuberculosis*.

In summary, we screened and identified a class of macrotetrolide products with antituberculosis action in this manuscript. Although we had initially revealed the mechanism of dinactin, the bactericidal effect of dinactin on *M. tuberculosis* dependent on the dissipation of PMF, disrupted redox homeostasis, and targeted the cell wall to synthesize associated membrane proteins, which may interact more easily with compounds from periplasmic space. Furthermore, the precise mechanism by which dinactin kills mycobacterial cells requires further investigation, and more in vivo efficacy and safety studies are required to further optimize its pharmacodynamic activity.

## Experimental methods

### Culture conditions of strains and eukaryotic cell lines

All mycobacteria strains and Mycobacterium abscessus were grown in Middlebrook 7H9 broth supplemented with 10% oleic albumin dextrose catalase (OADC) and 0.2% glycerol, 0.05% Tween 80 or on 7H11 agar plates (supplemented with 0.5% glycerol and 10% OADC). The *M. tuberculosis* clinical isolates used in this study were a gift from the Fifth People's Hospital of Heze. The other strains, including *Bacillus subtilis*, *Escherichia coli*, and *Klebsiella pneumonia* were grown in the Luria broth base. *Staphylococcus aureus*, methicillin-resistant *Staphylococcus aureus* (MRSA), *Streptococcus*, *Listeria monocytogenes*, and *Pseudomonas aeruginosa* were grown in brain heart infusion broth. Vero cells and THP-1 monocytes were grown in Dulbecco's modified Eagle's medium (DMEM) and RPMI-1640 medium supplemented with 10% fetal bovine serum at 37 °C with 5% $CO_2$, respectively. Ovine red blood cells were purchased from Bersee.

### Compound library and drugs used in this study

A library of 4406 unknown active natural products from plants and microorganisms (MEGxm and MEGxp) was obtained from AnalytiCon Discovery. Dinactin and its homologs were purchased from GlpBio. Other drugs used in this research were obtained from TargetMol.

### MIC and MBC determination

The minimum inhibitory concentration (MIC) of mycobacteria was determined using the resazurin reduction microplate assay (REMA)[57], with slight modifications. Briefly, mycobacterial cells grown to midexponential phase were diluted to OD600 = ~0.05 in medium and test compound solutions were diluted serially twofold. The plates were incubated for 7 days for *M. tuberculosis* H37Rv and clinical isolates (5 days for *M. bovis* BCG, 3 days for *M. abscessus* and *M. smegmatis* mc2155), then 30 µl of 0.1% resazurin solution was added to the cell wall, incubated overnight. A change from blue to pink indicates bacterial growth. Therefore, the MIC was defined as the lowest drug concentration that prevented color change. For other strains' susceptibility testing, the exponential phase of strains were diluted to OD600 of 0.001 and seeded into 96-well plates containing test compounds. After overnight incubation at 37 °C, the MIC was determined as the minimum concentration where no growth of strains could be detected by eye. All antibacterial experiments were carried out with at least two biological replicates.

The minimum bactericidal concentration (MBC) of mycobacteria was determined by incubating various concentrations of the test compound. After 7 days of incubation, the cultures were tenfold serially diluted and plated onto 7H11 agar plates to count the colony-forming units (CFU). The MBC is the compound concentration at which the number of bacteria has decreased by 2 logs compared to when the compound was first exposed.

### Cytotoxicity and hemolysis assay

The cytotoxicity of dinactin and its homologous against Vero cells and THP-1 cells were determined. Test compound solution was twofold serially diluted in cell medium to generate compound concentrations ranging from 128 to 1.0 µg/ml. WST-1 (Beyotime) solution was added to plated cells after 48 h incubation. After 2 h, the absorbance at 450 nm was detected using a TECAN Spark 10 M microplate reader. The IC50 (the minimal concentration of compound that inhibited viability by 50%) was extrapolated by measuring absorbance at 450 nm of the experiment group versus the untreated group.

Ovine red blood cells were used to determine the hemolysis of dinactin and its homologs. Briefly, Dinactin and its homologous were twofold serially diluted in PBS (pH 7.4). A lysis buffer (10 mM Tris, pH 7.8, 0.32 M sucrose, 5 mM $MgCl_2$, 10% Triton X-100) was used as a positive control. Hemocytes were added to drug dilutions and incubated at room temperature for 1 h. The samples were centrifuged at 500×*g* for 5 min. Finally, the supernatant (10 µl) was added to 90 µl of PBS to determine the absorbance at 450 nm (TECAN Spark 10 M).

### Bactericidal effect of dinactin on replicating and nutrient-deficient non-replicating mycobacteria

Log-phase *M. tuberculosis* H37Rv cultures were prepared in fresh medium to an OD600 of 0.05. The prepared cultures were treated with the desired concentrations of dinactin. Separately, the culture was incubated with 1 µg/ml of rifampicin as a positive control and 0.1% DMSO as a solvent control. At each time point, tenfold serially growing cultures of BCG were harvested by centrifugation, washed twice with PBS supplemented with 0.025% Tween 80, and resuspended the bacteria in the nutritionally deprived medium for 6 weeks at 37 °C. The bactericidal activity of drugs against the resulting nutrient-starved bacilli was determined by exposing each culture to compounds for 7 days at 37 °C and then measuring CFU by plating appropriate dilutions on 7H11 agar plates.

### Intracellular activity in THP-1

The intracellular activity was evaluated in THP-1 cells. Briefly, the cells were seeded into 24-well plates and stimulated for differentiation by PMA treatment for 48 h. Macrophages were infected with M. tuberculosis H37Rv at an MOI of 10 to quantify intracellular M. tuberculosis. After 4 h, the infected THP-1 cells were washed to remove extracellular bacteria and fresh medium containing compounds was added. Isoniazid (1 µg/ml) and 0.1% DMSO served as positive control and vehicle control, respectively. After 48 h, cells were washed and lysed with 0.025% Triton X-100, and the cells' lysates were plated onto 7H11 agar to calculate CFUs.

### Compound interaction was determined by using a REMA checkerboard assay

A REMA checkerboard was used to test for interaction between dinactin and approved antituberculosis antibiotics against *M. tuberculosis* H37Rv and drug-resistant strains. A twofold serial dilution of dinactin was added along the ordinate, and an equal volume of serial dilution of other antibiotics was added along the abscissa into a 96-well microplate in an 8-by-8 square format. Bacilli were diluted and added to the microplates and incubated at 37 °C. As described in the MIC test, resazurin solution was added to each well as a viability marker. The interaction of the two drugs was categorized based on the fractional inhibitory concentration index (FICI). FICI was calculated by use of the following formula: FICI = FICA (MIC of drug A combination/MIC of drug A alone) + FICB (MIC of drug B combination/MIC of drug B alone), FICI ≤0.5 indicated synergy; 0.5 < FICI ≤4 indicated additive; FICI >4 indicated antagonism[58].

### Galleria mellonella infection model

*G. mellonella* larvae were stored in wood shavings in the dark at room temperature. Before inoculation, the larvae surfaces were decontaminated using 75% ethanol and were injected into the hemocoel by a micro syringe in the last hind pleopod, with 10 µl of *M. tuberculosis* H37Rv ($2.5 \times 10^6$ CFU) and all the infected larvaes were randomly grouped. For drug treatment, a 10 µl dose of single drug or drug combination was injected similarly 3 days post-infection. The experiment group (*n* = 30 larvae) was treated with 10 mg/kg dinactin, rifampicin, isoniazid or their combination, and the control group (*n* = 30 larvae) was treated with PBS. And then, larvae were maintained in petri dishes containing a base layer of filter paper at 37 °C in the dark for 5 days. The treated larvae were monitored daily, and survival was recorded. Five days after treatment, larvae were washed twice in sterile PBS and homogenized for CFU counting.

### Determination of intracellular ion content

The intracellular ion content of BCG was determined as described previously[59], with slight modification. Briefly, bacteria were cultured in the 7H9 medium to midexponential phase, and cultures were collected and washed three times with deionized water. The pellets were resuspended in a 7H9 medium containing 0.25 µg/ml of dinactin, 0.1% DMSO and 0.25 µg/ml of valinomycin (a positive control). After 24 h, the samples were collected and weighed to ensure the pellets were of similar mass. Pellets were treated with lysozyme (20 mg/ml, dissolved in deionized water) for 24 h at 37 °C. Subsequently, cells were dissolved in Bugbuster (Novagen) according to the technical manual and incubated on a rotating mixer at a slow setting for 24 h. The BCA method measured the total protein for each sample according to the manufacturer's instructions. Each sample was diluted 100-fold in 2% molecular grade nitric acid to a total volume of 10 ml. Samples were analyzed by Inductively coupled plasma mass spectrometry (ICP-MS) (Agilent 7900).

### Bacterial membrane morphology, membrane permeability, and membrane fluidity assay

**Bacteria membrane morphology assay**. BCG was grown to OD600 = ~0.8 at 37 °C, and the culture was washed two times with PBS. Bacteria cultures were treated with 0.25 and 0.5 µg/ml of dinactin, 0.125 µg/ml of isoniazid, or 0.1% DMSO (control group) at 37 °C for 24 h. The treated bacteria samples were resuspended in 2.5% glutaraldehyde for 2 h at room temperature. Fixed cells were washed three times with 0.1 M phosphate buffer (pH 7.4) before being fixed for 2 h at room temperature in 0.1 M phosphate buffer (pH 7.4) containing 1% citric acid. The samples were washed three times with 0.1 M phosphate buffer (pH 7.4). Finally, cells were dehydrated with gradient absolute ethanol and isoamyl acetate. The dehydrated samples were dried in a critical point dryer (K850, Quorum). The dried samples were placed on a double-sided adhesive tape of a conductive carbon film and in an ion sputtering apparatus (MSP-2S, IXRF) for gold spraying. The samples were observed with a scanning electron microscope (SU8100, HITACHI).

**Bacteria membrane permeability assay**. Mid-exponential phase BCG cultures were washed three times with PBS and diluted to OD600 = ~0.1. The bacterial suspension was mixed with PBS containing twice the desired concentrations of compounds in black, clear-bottom, 96-well plates. SYTOX Green (Molecular Probes) was added to the bacterial suspension to a final concentration of 5 µM and incubated for 30 min at room temperature. Bacilli lysed with bead beating were used as a positive control, and 0.1% DMSO served as a vehicle control. Fluorescence was measured using a spectrophotometer (TECAN Spark 10 M) with excitation and emission wavelengths of 488 and 525 nm, respectively.

**Bacteria membrane fluidity assay**. Bacterial membrane fluidity was estimated by slightly modifying previously described methods[27]. First, log-phase BCG were diluted to OD600 of 0.05. Next, the diluted culture was incubated with 10 µM Laurdan (GlpBio) for 30 min at room temperature in the dark. The labeled bacteria were washed three times with PBS and resuspended in fresh medium, and the prestained culture was mixed with fresh medium containing twice the desired concentrations of compounds in a black, clear-bottom, 96-well plate. The membrane fluidizer benzyl alcohol (50 mM) was used as a positive control, and 0.1% DMSO served as a vehicle control. The plate was incubated for 3 h at 37 °C in the dark. The Laurdan fluorescence intensities were measured using a spectrophotometer (TECAN Spark 10 M) with emission wavelengths of 435 and 490 nm upon excitation at 350 nm. The Laurdan GP was calculated using the formula GP = (I435− I490)/(I435 + I490).

### Determination of membrane potential and electrochemical proton gradient

The transmembrane potential ($\Delta\Psi$) and electrochemical proton gradient ($\Delta$pH) were determined using a corresponding fluorescent probe. For $\Delta\Psi$, BCG cultures were washed with PBS and resuspended in fresh 7H9 medium. Then cultures were incubated with 30 µM DiOC2(3) (3,30-diethyloxacarbocyanine fluorescent iodine dye) for 30 min at room temperature. Stained cultures were subsequently washed with 7H9 and mixed with fresh medium containing twice the desired concentrations of compounds in black, clear-bottom, 96-well plates for 24 h. In addition, 5 µM of the protonophore carbonyl-cyanide 3-chlorophenylhydrazone (CCCP) as a positive control for membrane depolarization and DMSO was used as a vehicle control. At selected time points, the fluorescence of green (Ex488 nm/Em530 nm) and red (Ex488 nm/Em610 nm) was measured using a spectrofluorimeter (TECAN SPARK 10 M). Membrane potential was measured as a ratio of red to green fluorescence intensity. For $\Delta$pH, BCG culture was washed with HEPES (pH 7.0) and resuspended in the same buffer, and then prepared cultures were incubated with 10 µM BCECF-AM (a pH-sensitive fluorescence probe) at 37 °C. Prestained cultures were treated with the same concentrations described in the $\Delta\Psi$ assay, and the fluorescence intensity was determined at Ex488 nm and Em535 nm.

### Resazurin reduction assay

Resazurin reduction assay was estimated by slightly modifying previously described methods[60]. Exponential-phase BCG cultures were washed with PBS and diluted to an OD600 of 0.3 in fresh 7H9 broth. Bacteria cultures were treated with different compound concentrations of dinactin, 10 µM of CCCP or 0.1% DMSO (control group) at 37 °C. At 1, 2, and 3 h respectively, remove 1 mL from each flask, centrifuge to remove the supernatant, and resuspend the bacterial pellet in fresh medium, adjusting to an OD600 of 0.1, and subsequently add resazurin reagent to the diluted bacterial suspension to achieve a final concentration of 100 µg/mL. After thorough mixing, dispense the mixture into a 96-well plate with black transparent wells and incubate in an incubator for 10 min, and the fluorescence intensity was determined at Ex540 nm and Em630 nm.

### Determination of intracellular ATP content and NADH/NAD+ ratio

**ATP**. The intracellular ATP of BCG or non-replicating mycobacteria was quantified using the BacTiter-Glo microbial cell viability assay kit (Promega). Briefly, mycobacteria cultures were exposed to different compound concentrations for 24 h, and then cultures were collected and bedaquiline as a positive control. Treated mycobacteria lysates were mixed with the BacTiter-Glo reagent in a black, clear-bottom 96-well plate and then incubated for 5 min in the dark. The luminescence was measured using a Multimode Reader (TECAN SPARK 10 M) and was displayed as relative luminescence units (RLU).

**NADH/NAD +**. Exponential-phase BCG cultures were washed with PBS and resuspended in fresh 7H9 broth. NADH and NAD+ of mycobacteria were extracted as described previously[61], with appropriate modification. Briefly, prepared cultures were treated with certain concentrations of dinactin for 24 h. After then, cultures were harvested, and pellets were resuspended in 0.2 M NaOH (300 µl, NADH extraction) or 0.2 M HCl

(300 μl, NAD+ extraction), heated at 55 °C for 10 min and cooled to 0 °C. The extracted samples were neutralized by adding an equal volume of 0.1 M HCl (NADH extraction) or 0.1 M NaOH (NAD+ extraction). Cell debris was removed, and supernatants were transferred to a new tube. The concentration of NADH and NAD+ were detected using the NAD +/NADH assay kit (Beyotime) described in the manual. Firstly, 20 μl of the extracted supernatant prepared above and 90 μl alcohol dehydrogenase were added into a black, clear-bottom 96-well plate and incubated for 10 min at 37 °C in the dark. Subsequently, 10 μl reaction reagent was added and reacted for 45 min at 37 °C in the dark. The absorbance (OD450 nm) was determined using a spectrofluorimeter (TECAN SPARK 10 M). NADH standarized solutions ranging from 1 to 10 μM were used to establish a standard curve.

## Oxygen consumption and ROS determination

**Oxygen consumption measurement.** Methylene blue solution (0.1%, Sigma-Aldrich) was used as an oxygen indicator. Exponential-phase BCG cultures were prepared in fresh 7H9 medium and five-millilter screw-cap glass vials were filled with 4.5 ml of mycobacterial culture (OD600 = ~0.3) in the presence of the desired concentrations of compound and the cultures were treated with the test compound for 6 h at 37 °C. Thioridazine (THD, 80 μM) was used as the positive control, inhibiting respiration and blocking oxygen consumption. After adding methylene blue (0.001%), all screw-cap vials were placed and incubated at 37 °C in an anaerobic jar (MGC). Oxygen was removed by an anaerobe sachet (MGC AnaeroPack). The vials were removed from the anaerobic jar after 120 h for image recording.

**ROS measurement.** Log phase BCG cultures were washed and resuspended in fresh 7H9 broth to obtain an OD600 = ~0.5. About 10 μM DCFH-DA was added to the bacterial suspension and incubated at 37 °C for 30 min. Labeled cultures were washed with PBS. Then the bacterial suspension was added to a black 96-well plate and mixed with dinactin solution for 24 h, with $H_2O_2$ as a positive control. After incubation for a specific time, the fluorescence value (Ex492 nm/Em 525 nm) was determined using a spectrofluorimeter (TECAN SPARK 10 M). The experiment was performed with three technical replicates and independently repeated at least once.

## Isolation and sequencing of resistant mutants

BCG-resistant mutants were generated by plating $1 × 10^9$ CFU bacteria from log-phage culture on solid agar containing 0.5, 1, and 2 μg/ml of dinactin and incubated for 4–5 weeks at 37 °C. Potentially resistant colonies were inoculated into 7H9 media containing 0.5 μg/ml of dinactin to confirm phenotypic resistance and subjected to MIC determination. The genomic DNA of these resistant mutants and the wild-type strain were prepared for whole-genome sequencing.

## Molecular docking

To determine the binding mode of dinactin and cpsA, we performed a molecular docking analysis between dinactin and the protein using Auto-Dock Vina. Due to cpsA shared only between 21 and 29% identity with other LCP proteins from *S. aureus*, *Streptococcus pneumonia*, and *B. subtilis*[42]. Thus, we utilized an AlphaFold structure prediction of cpsA (model ID: AF-O06347-F1) to proceed with the docking procedure[43]. Dinactin molecular structure was optimized and then docked with the specific region of the protein. Here, we created a grid box of dimension $5 × 3 × 8$ Å along the direction of XYZ with a spacing of 0.375 Å. Lamarck Genetic Algorithm parameters were set to maximum efficiency values to increase prediction efficiency. Based on the docking score, the model with the highest score was chosen for further analysis.

## Protein expression and purification

As described previously[42], the gene segment is devoid of the N-terminal transmembrane domain, CpsA1ΔTM (corresponding to residues 38–498) and cpsAΔTM (corresponding to residues 49–512). The segments were cloned into the PET28a vector and recombinant plasmids were transformed into *E. coli* BL21 (DE3). The mutant plasmid (PET28a-cpsAAsp347-GlyΔTM) was generated using the Fast Mutagenesis

System kit based on a wild-type construct. The recombinant expression system was incubated in LB medium (plus kanamycin) and induced by 1 mM IPTG at 16 °C for 20 h. Bacterial cells were centrifuged and resuspended in lysis buffer (50 mM Tris-HCl, 500 Mm NaCl, and 10% glycerol, pH 8.0). The lysates were centrifuged and bound to Ni-agarose affinity resin, eluted with a buffer containing 50 mM Tris, 5% glycerol, and 100 mM imidazole, pH 8.0. The purified protein was ultrafiltered, concentrated, and stored at −80 °C.

## Microscale thermophoresis (MST) binding assay

MST was conducted using an NT.115 MST instrument (Nano Temper Technologies GmbH). Purified protein was labeled with the Monolith NTTM Protein Labelling Kit RED (Nano Temper). Labeled protein was diluted to 20 nM. Dinactin was prepared in dilutions ranging from 654 μM to 20 nM. All samples were diluted in a buffer containing PBS (pH 7.4) and 0.05 (v/v)% Tween-20. Subsequently, samples were incubated for 30 min in the dark and loaded into the customized capillaries and the data were recorded. The dissociation constant (*K*d) values were analyzed using the NTAnalysis software.

## Construction of cpsA knock-out mutant and overexpression strains

Mycobacteriaphage-based specialized transduction was used to knockout the target gene in BCG[62]. The construction of cpsA knock-out mutants of BCG involved amplifying the upstream and downstream homologous arms by using the primer pairs cpsA-LFP (5'-TTTTTTTTTTCCATAAATTGGT TCCACGGCCTGCCGGTTTT-3') and cpsA-LRP (5'-TTTTTTTTTCCAT TTCTTGGATTGCCCTCAGAACGCGCCA-3'), cpsA-RFP (5'-TTTTT TTTCCATAGATTGGGGCGTGCCCTGCGTGAACTA-3') and cpsA-RRP (5'-TTTTTTTTTCCATCTTTTGGACCGGAGGTGGCAGCGGAA T-3'), respectively. The two homologous arms and the pYUB1471 plasmid were digested with Van91I and ligated with T4 DNA ligase. The recombinant plasmid pYUB1471 and temperature-sensitive phage phAE159 were digested with PacI and dephosphorylated using the thermal alkaline phosphatase kit. Subsequently, the products were ligated using T4 DNA ligase to construct a shuttle plasmid phAE159-cpsA and packed with MaxPlax Lambda Packaging Extracts (Epicenter Biotechnologies). The packed shuttle plasmid was transformed into E.coli HB101 competent cells. High-titer phages were prepared by electrotransformation of the corrected recombinant phAE159 plasmid into M. smegmatis mc2155 at 30 °C. Lastly, BCG was infected with high-titer phage lysate at MOI of 1:1 and 10:1 at 37 °C and the deletion mutant was confirmed by utilizing the primer pairs cpsA-in-F (5'-GGCTACAACACGAACACGCTGATA-3') and cpsA-in-R (5'-GAATGCTTGCTGGCGGTGG-3'), cpsA-out-F (5'-ATCTCGTCAGA CACCTAACCCGCTAA-3') and cpsA-out-R (5'-TCTGGTTGTGCCC GTCACAGAGT-3'). The complemented cpsA strain was generated by transforming pMV261-cpsA into a knockout strain. The cpsA or CpsA1 gene was amplified by PCR from the wildtype with the primer pairs cpsA FP (5'-CCGGAATTCATGGCGCGTTCTGAGGG-3'), cpsA RP (5'-CCCAA GCTTCTAGTTCACGCAGGGCACG-3'), CpsA1 FP (5'-CCGGAATTC GTGATGTCTGCGCAACGTGT-3') and CpsA1 RP (5'-CCCAAGCTT TCAGT TGATGCACTCCGGCG-3'). The PCR product was digested with the appropriate restriction enzymes and cloned into the pMV261 vector to yield pMV261-cpsA or pMV261-CpsA1. The vector and sequence-verified plasmids were transferred into BCG to generate strains of BCG::pMV261, BCG::pMV261-cpsA, and BCG::pMV261-CpsA1.

## Statistical analysis and reproducibility

Each treatment was triplicated, and the experiments were repeated at least twice to ensure reproducibility. The data were analyzed using GraphPad Prism software 8.0 or Excel. Unpaired student *T*-tests were used to analyze

the differences between treated and control groups. All statistical data were represented as mean ± SD. Statistical significance was defined as $*P \le 0.05$, $**P \le 0.01$, and $***P \le 0.001$.

## Reporting summary

Further information on research design is available in the Nature Portfolio Reporting Summary linked to this article.

## Data availability

All data supporting the findings of this study are available in the manuscript (Supporting Information and Supplementary Data). Source data underlying graphs can be obtained from Supplementary Data 1. The sequence data can be obtained from the Sequence Read Archive (SRA) (The accession codes: SRR36947364, SRR36947365, SRR36947366, SRR36947367, SRR36947368, and SRR36947369). If there are any special requests or questions for the data, please contact the corresponding author (Chen Tan).

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

## Acknowledgements

This work was supported by the National Key R&D Program [2021YFD1800402] and the National Natural Science Foundation of China [32273008].

## Author contributions

Conceptualization, C.T., X.W., and H.C.; Screen development and performed experiment, G.W., W.D., Y.B., and Y.L.; G.W., W.D., and J.T. analyzed data; MST assay, H.L. and W.L.; Molecular docking, C.W.; Knockout mutant strains, P.L. and R.W.; Manuscript writing, G.W.; Manuscript review and editing, all authors; Funding acquisition, C.T.

## Competing interests

The authors declare no competing interests.
