## [Transparent Peer Review file · Communications Biology]

Naturally occurring dinactin targets cpsA protein and kills Mycobacterium tuberculosis by disrupting proton motive force

Corresponding Author: Professor Chen Tan

Version 0:

Reviewer comments:

Reviewer #1

(Remarks to the Author)

In this manuscript, the authors describe their characterization of the anti-mycobacterial activity of dinactin, a macrotetrolide they identified from a screen of a natural products library. Dinactin displayed synergy with several anti-tubercular drugs, which through results of a variety of assays was attributed to disruption of membrane integrity and proton motive force. The authors go on to demonstrate antitubercular activity in phagocytes and a wax moth larvae infection model. After isolation and whole genome sequencing of resistant mutants, the authors identify CpsA, which is involved in ligating arabinogalactan to peptidoglycan in the cell wall, as a target. The data convincingly supports that dinactin acts by interfering with membrane function and interacts with CpsA to compromise growth, but in several places the manuscript lacks key details for the interpretation of the data, overinterprets what can be concluded from the data, and neglects to follow up on an important finding. Overall, the manuscript is adequately written, and the experiments are sufficiently well described with necessary controls, but frequent typographical errors and missing words or unusual phrasings suggest it would substantially benefit from careful editing. The most critical issues below are L264 and L440.

Specific comments:

L51 – isoniazide is an unusual spelling, which varies from the more common isoniazid used elsewhere (L172). Use consistent spelling, preferably isoniazid.

L89 – metagenom

L102 – combiing

L110 – Despite -> Although

L117 – ionophore which disrupts

Fig 1E – missing Y axis label

Table S1 – check extreme rifampicin susceptibility of BCG

L183 – Dinactin and Isoniazid not at the same concentration

L210 – activity

Fig 2A-H – these are low information panels, better presented by table S3.

Fig 2H – PBTZ169, not PBTZ168

Fig 2F & 2G – same Y axis label

Fig 2I & 2J- these are low information panels, better presented by table S4 and the color of the squares provides no information.

L254 – CFU drop

L264 – It seems unlikely that the accumulation experiment, with detection by UV, would be able to unambiguously distinguish and measure the drugs amongst the thousands of metabolites in the cell. Is it possible this was actually done by mass spectrometry? If not, please provide chromatograms demonstrating that the drugs can be specifically measured with labels to indicate the peaks, both in the standards run and the extracts.

L288 – diaplayed

L383 – pronon

Fig 4A, B, E, I, J – it is difficult if not impossible to identify the time treatment in of each of these assays, either in the figure, the legend, the main text, or the materials and methods. Add where missing, preferably so it can be seen when reading the text or legend.

Fig 4B – isoniazid images missing

L405 – wild-type – persisters are not genetic variants, wild-type is not appropriate here, maybe use “actively growing cultures”

L432 – spontaneously resistant

L440 – this analysis identifies not just *cpsA* but also the “putative protein” variant at position 2203278. Comparing the various mutants suggest that whereas the *cpsA* variant may account for a 4-fold increase in MIC, the 2203278 variant likely accounts for 8-fold. It is surprising that this mutation was not followed up with genetic validation as it is likely more important to the activity of dynactin than *CpsA*.

L460 – the target -> a target

L473 – the target -> a target

L524 – occur -> express

L528 – the target -> a target

L575 – infectious mouse -> mouse infection

L578, 582 – spontaneously resistant

L588, 589 – this does not follow, this should be something like – one of these potential targets is the key ligase responsible for...however it remains to be clearly identified.

Refs 57-61 are not in the main text, are repeated in the supplementary

In supplementary:

Eukaryocyte growth -> viability

Log phage -> log phase

Check SYTOX green assay, cells should not have been pre-stained.

Method for figure 4H is missing.

Reviewer #2

(Remarks to the Author)

The manuscript by Wang et al describes the testing of dynactin against *Mycobacterium tuberculosis* (Mtb) and *Mycobacterium bovis* BCG. The authors used MIC, various in vitro and in vivo testing conditions, mutant isolation, and biochemical analyses to decipher the activity and mode of action of dynactin against mycobacterial species. This is an interesting study that requires additional information and critical modifications.

The following points need to be addressed:

1. The authors' findings regarding *cpsA* and dynactin resistance present an interesting paradox. While the isolation of dynactin-resistant BCG mutants all showing the same mutation in *cpsA* initially suggests it could be a target, the limited two-fold increase in MIC upon *cpsA* overexpression in wild-type BCG casts doubt. Typically, significant overexpression of a direct drug target leads to a much more substantial increase in MIC, often by 10-fold or more. The modest two-fold increase observed here might not be relevant and suggests *cpsA* may not be the primary or sole target of dynactin. This leads to the following question: did any of the highly dynactin-resistant clinical isolates, some showing up to an 8x MIC increase, possess mutations in *cpsA*? Investigating the genomes of these highly resistant clinical isolates could provide valuable insights. It could be more informative to identify what specific mutations or genetic changes are responsible for their high level of dynactin resistance, as this could point to novel, more relevant drug targets or resistance mechanisms beyond *cpsA*.
2. Furthermore, the authors did their biochemical studies using *M. bovis* BCG. Their BCG strain was 4 times more susceptible to dynactin than the parental strain *M. bovis* or *M. tuberculosis* (Mtb). What is the genetic component of this increased susceptibility?
3. The authors tested dynactin against non-replicating *M. bovis* BCG (Figure 1F) and compared the dynactin result to isoniazid (INH) in this condition. The authors' use of isoniazid (INH) as a positive control when testing dynactin against non-replicating *M. bovis* BCG (Figure 1F) is puzzling. It's widely established that INH has no activity in non-replicating bacterial states. Therefore, INH effectively served as a negative control in this experiment, not a positive one, as stated by the authors. For a positive control, the authors should have used an antibiotic known to be active against non-replicating mycobacteria, such as rifampicin. This would have provided a more accurate and meaningful comparison for dynactin's efficacy.
4. Figure 1G/line 200: The authors claim that dynactin achieves a 2-log killing of *M. tuberculosis* (Mtb) within the THP-1 cell line. To substantiate this claim, it is crucial to establish the baseline bacterial load. This 2-log reduction is only accurately quantifiable if the comparison is made between the initial inoculum (colony-forming units, or CFUs, at the beginning of the infection, i.e., Day 0) and the CFUs observed at Day 2 in the presence of dynactin. Currently, Figure 1G only presents the DMSO control titer at Day 2. To properly evaluate dynactin's killing efficacy in THP-1 cell lines, the CFU count at Day 0 (the inoculum) must be included in Figure 1G. Without this critical baseline, the asserted 2-log killing cannot be definitively verified.
5. Figure 3D: add the inoculum size (CFUs at day 0) to the figure to better estimate the efficacy of each treatment after 5 days.
6. The authors showed that the NADH/NAD⁺ ratio increased upon dynactin treatment. Was the increase in this ratio due to an increase in NADH concentration or a decrease in NAD⁺ concentration?
7. Line 171: Figure 1E shows a 5-6 logs reduction in CFUs with 4 mg/l and 8 mg/l of dynactin, not 2 mg/l and 4 mg/l as stated in the text.
8. Line 197: The testing of dynactin in the THP1 cell line is an in vitro model of Mtb infection, not an ex-vivo model.
9. Figures 2F and 2G have the same y axis legend.
10. Lines 259-268: add UPLC chromatograms to supplemental data.

Reviewer #3

(Remarks to the Author)

I co-reviewed this manuscript with one of the reviewers who provided the listed reports. This is part of the Communications Biology initiative to facilitate training in peer review and to provide appropriate recognition for Early Career Researchers who co-review manuscripts.

Version 1:

Reviewer comments:

Reviewer #1

(Remarks to the Author)

This reviewer appreciates the efforts that have been made by the authors to respond to the previous critiques. Several issues remain incompletely addressed which could significantly improve the readability of the paper and confidence in the results reported.

Table S1 - though the authors state that the significant sensitivity of BCG to rifampicin (15x greater than H37Rv, 30x greater than their *M. bovis* strain) was reproducible, it brings into question the genetic background of this strain that plays a key role in the paper. If it is not dealt with in this paper, it should be explored separately. Whatever variant is responsible could make an exciting drug target.

Figures 2A-J - it was not clear from Communications Biology's instructions to authors that the authors are limited to 2 tables. It would be worth checking with the editors as these figure panels can be much more efficiently presented as tables.

Figure S2 - though the chromatograms are a welcome addition, this data remains unconvincing, particularly for rifampicin, which implies a 40% difference in cell number used for dinactin-treated vs. DMSO-treated samples if correct. As this data is not crucial to the conclusion of the paper it would probably be best to leave it out.

Figure 4A, B, E, I, J - it would be easier for the reader if the time of treatment was in the figure legend rather than buried in the supplementary

Supplementary L62 - says isoniazid although this control has been changed to rifampicin

Supplementart L192 -the timing here is not clear, 72 hour treatment is clear, 10 minute incubation after resasuzrin is clear, but what what is happening for the 1, 2, and 3 hours? The 10 minutes with resazurin seems very fast, usually incubation for MIC are overnight.

Supplementary overall - this would benefit from careful proofreading. Isoniazid is again spelled here in two different ways and there are numerous misspellings throughout.

Reviewer #2

(Remarks to the Author)

The reviewer thanks the authors for their carefully crafted responses.

Two points still need to be addressed by the authors:

Point # 1: Since the discussion states that "dinactin might have other targets or pathways to kill mycobacteria (lines 585-586), line 60 in the abstract, which states that *cpsA* is the principal target of dinactin, should be modified.

The authors' response to the review's point #2 is insufficient because the premise that BCG is more susceptible to antibiotics than *Mtb* is incorrect. Far from being more susceptible, BCG exhibits similar or higher MICs against multiple mycobacterial antibiotics. Therefore, the fourfold increase in BCG's susceptibility to dinactin relative to *M. bovis* and *Mtb* remains unexplained.

Reviewer #3

(Remarks to the Author)

I co-reviewed this manuscript with one of the reviewers who provided the listed reports. This is part of the Communications Biology initiative to facilitate training in peer review and to provide appropriate recognition for Early Career Researchers who co-review manuscripts.

Version 2:

Reviewer comments:

Reviewer #1

(Remarks to the Author)

The revisions have addressed the major concerns. The point shared by the reviewers that the investigators' BCG is unusually sensitive to rifampicin remains, despite some of the offered explanations. In separate work, it would be worthwhile for the authors to whole genome sequence their BCG and compare it to the most closely related isolate that lacks this profound sensitivity.

Reviewer #2

(Remarks to the Author)

No further comments.

Reviewer #3

(Remarks to the Author)

I co-reviewed this manuscript with one of the reviewers who provided the listed reports. This is part of the Communications Biology initiative to facilitate training in peer review and to provide appropriate recognition for Early Career Researchers who co-review manuscripts.

Reviewer #1 (Remarks to the Author):

In this manuscript, the authors describe their characterization of the anti-mycobacterial activity of dinactin, a macrotetrolide they identified from a screen of a natural products library. Dinactin displayed synergy with several anti-tubercular drugs, which through results of a variety of assays was attributed to disruption of membrane integrity and proton motive force. The authors go on to demonstrate antitubercular activity in phagocytes and a wax moth larvae infection model. After isolation and whole genome sequencing of resistant mutants, the authors identify CpsA, which is involved in ligating arabinogalactan to peptidoglycan in the cell wall, as a target. The data convincingly supports that dinactin acts by interfering with membrane function and interacts with CpsA to compromise growth, but in several places the manuscript lacks key details for the interpretation of the data, overinterprets what can be concluded from the data, and neglects to follow up on an important finding. Overall, the manuscript is adequately written, and the experiments are sufficiently well described with necessary controls, but frequent typographical errors and missing words or unusual phrasings suggest it would substantially benefit from careful editing. The most critical issues below are L264 and L440.

Specific comments:

1. L51 – isoniazide is an unusual spelling, which varies from the more common isoniazid used elsewhere (L172). Use consistent spelling, preferably isoniazid.

We thank the reviewer for identifying this error and have corrected this throughout the manuscript.

2. L89 – metagenom

We thank the reviewer for identifying this error and have made correction in lines 89 .

3. L102 – combiing

We thank the reviewer for identifying this error and have made correction in line 102.

4. L110 – Despite -> Although

We thank the reviewer for identifying this error and have corrected this in line 110.

5. L117 – ionophore which disrupts

We thank the reviewer for identifying this error and have corrected this in line 117.

6. Fig 1E – missing Y axis label

We thank the reviewer for identifying this error and have made correction in Fig.1E.

7. Table S1 – check extreme rifampicin susceptibility of BCG

We are grateful for the reviewer's comments and we have re-tested and confirmed the MIC of rifampicin, the result is consistent with our previous data.

8. L183 – Dinactin and Isoniazid not at the same concentration

Thank you for the comment, we determine the dosage concentration based on the MIC of the drug, whether it's dinactin or isoniazid, the drug concentration is set as 16 X MIC. The MIC values of drug are shown in Table 1.

9. L210 – activitiy

We thank the reviewer for identifying this error and have corrected in line 210.

10. Fig 2A-H – these are low information panels, better presented by table S3.

We are very grateful to the reviewers for their suggestions on our manuscript, and we apologise for any distress, but given the article's limitations on the number of tables, we

have had to make graphs of the data in some of the tables, and instead display the data tables in the supplementary material.

11. Fig 2H – PBTZ169, not PBTZ168

We thank the reviewer for identifying this error and have corrected in figure 2H.

12. Fig 2F & 2G – same Y axis label

We thank the reviewer for identifying this error and have corrected in figure 2F and 2G.

13. Fig 2I & 2J- these are low information panels, better presented by table S4 and the color of the squares provides no information.

We are very grateful to the reviewers for their suggestions on our manuscript, and we apologise for any distress, but given the article's limitations on the number of tables, we have had to make graphs of the data in some of the tables, and instead display the data tables in the supplementary material.

14. L254 – CFU drop

We thank the reviewer for identifying this error and have corrected in line 254.

15. L264 – It seems unlikely that the accumulation experiment, with detection by UV, would be able to unambiguously distinguish and measure the drugs amongst the thousands of metabolites in the cell. Is it possible this was actually done by mass spectrometry? If not, please provide chromatograms demonstrating that the drugs can be specifically measured with labels to indicate the peaks, both in the standards run and the extracts.

We appreciate the reviewer's comments and we have already included the corresponding ULPC chromatogram data in the supplementary materials in Figure S2.

16. L288 – displayed

We thank the reviewer for identifying this error and have corrected in line 287.

17. L383 – pronon

We thank the reviewer for identifying this error and have corrected in line 382.

18. Fig 4A, B, E, I, J – it is difficult if not impossible to identify the time treatment in of each of these assays, either in the figure, the legend, the main text, or the materials and methods. Add where missing, preferably so it can be seen when reading the text or legend.

We are grateful for the reviewers' suggestions and have added the specific processing times for the relevant experiments in supplementary materials in line 126, 140, 178, 211 and 239.

19. Fig 4B – isoniazid images missing

I apologise for troubling you with our unprecise experimental design, because in this experiment we did not design a SEM control test for isoniazid.

20. L405 – wild-type – persisters are not genetic variants, wild-type is not appropriate here, maybe use “actively growing cultures”

We thank the reviewer's comments and have corrected this in line 405.

21. L432 – spontaneously resistant

We thank the reviewer for identifying this error and have corrected in line 431.

22. L440 – this analysis identifies not just cpsA but also the “putative protein” variant at position 2203278. Comparing the various mutants suggest that whereas the cpsA variant may account for a 4-fold increase in MIC, the 2203278 variant likely accounts for 8-fold. It is surprising that this mutation was not followed up with genetic validation as it is likely more important to the activity of dinactin than CpsA.

We appreciate the reviewer's comments. In this assay, the process of inducing bacterial mutations through drug-induced stress is inherently random. Consequently, prior to commencing the experiment, we anticipated that sites exhibiting higher mutation frequencies within the acquired mutant strains would warrant our primary attention. Given that 3881452 (cpsA) harboured point mutations in the vast majority of mutant strains, we therefore focused our investigation on the function of the cpsA gene. As noted by the reviewers, our results also identify additional mutation sites that may influence dinactin's pharmacological activity. Although our manuscript did not investigate the role of the 2203278 mutation site in Mycobacterium tuberculosis resistance to dinactin, we believe this provides a crucial avenue for future research and analysis of dinactin's antibacterial mechanism and this is a gene that was not sufficiently addressed in our earlier investigations.

23. L460 – the target -> a target

We thank the reviewer for identifying this error and have corrected in line 459.

24. L473 – the target -> a target

We thank the reviewer for identifying this error and have corrected in line 472.

25. L524 – occur -> express

We thank the reviewer for identifying this error and have corrected in line 524.

26. L528 – the target -> a target

We thank the reviewer for identifying this error and have corrected in line 527.

27. L575 – infectious mouse -> mouse infection

We thank the reviewer for identifying this error and have corrected in line 574.

28. L578, 582 – spontaneously resistant

We thank the reviewer for identifying this error and have corrected in line 577, 581.

29. L588, 589 – this does not follow, this should be something like – one of these potential targets is the key ligase responsible for...however it remains to be clearly identified.

We appreciate the reviewer's comments and this statement was indeed not sufficiently rigorous. Following your advice, we have revised it in line 591-594.

30. Refs 57-61 are not in the main text, are repeated in the supplementary

We thank the reviewer's comments and have corrected in manuscript and supplementary materials.

In supplementary:

31. Eukaryocyte growth -> viability

We thank the reviewer's comments and have corrected in supplementary materials in line 48.

32. Log phage -> log phase

We thank the reviewer's comments and have corrected in supplementary materials in line 60.

33. Check SYTOX green assay, cells should not have been pre-stained.

We thank the reviewer's comments and have corrected in supplementary materials in line 150-158.

34. Method for figure 4H is missing.

We thank the reviewer's comments and have added the method in supplementary materials in line 189-199.

Reviewer #2 (Remarks to the Author):

The manuscript by Wang et al describes the testing of dynactin against *Mycobacterium tuberculosis* (Mtb) and *Mycobacterium bovis* BCG. The authors used MIC, various in vitro and in vivo testing conditions, mutant isolation, and biochemical analyses to decipher the activity and mode of action of dynactin against mycobacterial species. This is an interesting study that requires additional information and critical modifications.

The following points need to be addressed:

1. The authors' findings regarding *cpsA* and dynactin resistance present an interesting paradox. While the isolation of dynactin-resistant BCG mutants all showing the same mutation in *cpsA* initially suggests it could be a target, the limited two-fold increase in MIC upon *cpsA* overexpression in wild-type BCG casts doubt. Typically, significant overexpression of a direct drug target leads to a much more substantial increase in MIC, often by 10-fold or more. The modest two-fold increase observed here might not be relevant and suggests *cpsA* may not be the primary or sole target of dynactin. This leads to the following question: did any of the highly dynactin-resistant clinical isolates, some showing up to an 8x MIC increase, possess mutations in *cpsA*? Investigating the genomes of these highly resistant clinical isolates could provide valuable insights. It could be more informative to identify what specific mutations or genetic changes are responsible for their high level of dynactin resistance, as this could point to novel, more relevant drug targets or resistance mechanisms beyond *cpsA*.

*We appreciate the reviewer's comments. In this assay, the process of inducing bacterial mutations through drug-induced stress is inherently random. Consequently, prior to commencing the experiment, we anticipated that sites exhibiting higher mutation frequencies within the acquired mutant strains would warrant our primary attention. Given that *cpsA* harboured point mutations in the vast majority of mutant strains, we therefore focused our investigation on the function of the *cpsA* gene. This is why we selected *cpsA* for further investigation. Based on our research data, we understand the reviewer's point that *cpsA* may not be the primary target or sole target of dynactin. We have also described the potential causes of this situation in the discussion section of the manuscript in line 583-605.*

*Besides, our results also identify additional mutation sites that may influence dynactin's pharmacological activity, for example, the the 2203278 mutation site was observed in both of the two mutant strains. Although we did not further investigate the role of the 2203278 mutation site in *Mycobacterium tuberculosis* resistance to dynactin in our manuscript, we believe this provides a crucial avenue for future research and analysis of dynactin's antibacterial mechanism. Regrettably, the gene was not sufficiently research in our earlier investigations.*

2. Furthermore, the authors did their biochemical studies using *M. bovis* BCG. Their BCG strain was 4 times more susceptible to dynactin than the parental strain *M. bovis* or *M. tuberculosis* (Mtb). What is the genetic component of this increased susceptibility?

*We appreciate the reviewer's comments. BCG is an attenuated strain of *Mycobacterium bovis*, and the genome of BCG has significant differences compared to *M. bovis* or *M. tuberculosis*, with deletion of the RD1 region of the genome, which may affect the structural*

integrity and permeability of the BCG cell wall. Especially in the cell wall structure of M. tuberculosis, mutations may occur in the synthesis or assembly of lipids (such as mycolic acid, Lipidarabinomannan, etc.) in the BCG cell wall. This results in a lipid composition distinct from that of M. bovis and M. tuberculosis, conferring greater permeability to the BCG cell wall. Consequently, BCG exhibits heightened susceptibility to antibiotics.

3. The authors tested dynactin against non-replicating M. bovis BCG (Figure 1F) and compared the dynactin result to isoniazid (INH) in this condition. The authors' use of isoniazid (INH) as a positive control when testing dynactin against non-replicating M. bovis BCG (Figure 1F) is puzzling. It's widely established that INH has no activity in non-replicating bacterial states. Therefore, INH effectively served as a negative control in this experiment, not a positive one, as stated by the authors. For a positive control, the authors should have used an antibiotic known to be active against non-replicating mycobacteria, such as rifampicin. This would have provided a more accurate and meaningful comparison for dynactin's efficacy.

We appreciate the reviewer's comments. We have used rifampicin as a positive control in this assay in figure 1F.

4. Figure 1G/line 200: The authors claim that dynactin achieves a 2-log killing of M. tuberculosis (Mtb) within the THP-1 cell line. To substantiate this claim, it is crucial to establish the baseline bacterial load. This 2-log reduction is only accurately quantifiable if the comparison is made between the initial inoculum (colony-forming units, or CFUs, at the beginning of the infection, i.e., Day 0) and the CFUs observed at Day 2 in the presence of dynactin. Currently, Figure 1G only presents the DMSO control titer at Day 2. To properly evaluate dynactin's killing efficacy in THP-1 cell lines, the CFU count at Day 0 (the inoculum) must be included in Figure 1G. Without this critical baseline, the asserted 2-log killing cannot be definitively verified.

We thank the reviewer for this comments and have added the CFUs at day 0 in figure 1G

5. Figure 3D: add the inoculum size (CFUs at day 0) to the figure to better estimate the efficacy of each treatment after 5 days.

We appreciate the reviewer's comments and we have added the CFUs at day 0 in figure 3D.

6. The authors showed that the NADH/NAD⁺ ratio increased upon dynactin treatment. Was the increase in this ratio due to an increase in NADH concentration or a decrease in NAD⁺ concentration?

We thank the reviewer for this comment and our results shown that the concentration of NADH was significantly increase, meanwhile, the concentration of NAD⁺ also displayed a decreasing trend, but it is not in a concentration-dependent manner. I think the increase in the NADH/NAD⁺ ratio due to the increase in NADH concentration.

7. Line 171: Figure 1E shows a 5-6 logs reduction in CFUs with 4 mg/l and 8 mg/l of dynactin, not 2 mg/l and 4 mg/l as stated in the text.

We appreciate the reviewer's comments and have corrected in line 171.

8. Line 197: The testing of dynactin in the THP1 cell line is an in vitro model of Mtb infection, not an ex-vivo model.

We appreciate the reviewer's comments and have corrected in line 197.

9. Figures 2F and 2G have the same y axis legend.

We appreciate the reviewer's comments and have corrected this error in Figures 2.

10. Lines 259-268: add UPLC chromatograms to supplemental data.

We appreciate the reviewer's comments and we have already included the corresponding ULPC chromatogram data in the supplementary materials in Figure S2.

Reviewer #3 (Remarks to the Author):

I co-reviewed this manuscript with one of the reviewers who provided the listed reports. This is part of the Communications Biology initiative to facilitate training in peer review and to provide appropriate recognition for Early Career Researchers who co-review manuscripts.

We are grateful to the reviewers for their suggestions on our manuscript, which will assist us in better completing our research.

Reviewer #2 (Remarks to the Author):

The reviewer thanks the authors for their carefully crafted responses.

Two points still need to be addressed by the authors:

1. Point # 1: Since the discussion states that “dinactin might have other targets or pathways to kill mycobacteria (lines 585-586), line 60 in the abstract, which states that *cpsA* is the principal target of dinactin, should be modified.

We appreciate the reviewer’s comments and have revised our statement in line 59-61 in the abstract.

2. The authors’ response to the review's point #2 is insufficient because the premise that BCG is more susceptible to antibiotics than Mtb is incorrect. Far from being more susceptible, BCG exhibits similar or higher MICs against multiple mycobacterial antibiotics. Therefore, the fourfold increase in BCG’s susceptibility to dinactin relative to *M. bovis* and Mtb remains unexplained.

We appreciate the reviewer’s comments and we also understand the concerns of the reviewer: In our study, we think that the difference in antibiotic susceptibility between the BCG and H37RV strains is attributable to mutations in certain efflux pump genes within the bacteria, resulting in impaired efflux pump function. Behr et al. systematically identified multiple RD regions (such as RD1, RD2, etc.) absent in BCG relative to the Mycobacterium tuberculosis H37Rv strain through genomic comparison for the first time. The absence of these regions constitutes the molecular basis for BCG's attenuated virulence (Behr, M. A., et al. (1999). Comparative genomics of BCG vaccines by whole-genome DNA microarray. Science, 284(5419), 1520-1523.). Although no completely efflux pump genes are missing in BCG in the RD regions, several studies have identified specific gene mutations related to drug efflux and sensitivity, such as, Danilchanka et al. and Ramón-García et al. have indicated that the Rv1410c gene encodes a crucial efflux pump associated with drug resistance and cell envelope integrity in mycobacteria. Within BCG strains, a single nucleotide polymorphism (SNP) exists within the Rv1410c gene, resulting in an altered protein sequence (A202E). This mutation has been demonstrated to reduce the efflux pump's functionality, rendering BCG more susceptible to certain drugs (such as streptomycin and chloramphenicol) than H37Rv (Danilchanka, O., et al. (2008). Identification of a novel multidrug efflux pump of Mycobacterium tuberculosis. Antimicrobial Agents and Chemotherapy, 52(7), 2503-2511; Ramón-García, S., et al. (2011). The Mycobacterium tuberculosis efflux pump Rv1410c confers resistance to a diverse range of antituberculosis drugs. Antimicrobial Agents and Chemotherapy, 55(5), 2294-2297.). Therefore, we speculate that in BCG strains, dysfunction of the efflux pump may lead to reduced antibiotic efflux activity, thereby rendering BCG more susceptible to certain antibiotics. We wish that the above explanations will dispel your doubts.

Reviewer #1 (Remarks to the Author):

This reviewer appreciates the efforts that have been made by the authors to respond to the previous critiques. Several issues remain incompletely addressed which could significantly improve the readability of the paper and confidence in the results reported.

1. Table S1 - though the authors state that the significant sensitivity of BCG to rifampicin (15x greater than H37Rv, 30x greater than their M. bovis strain) was reproducible, it brings into question the genetic background of this strain that plays a key role in the paper. If it is not dealt with in this paper, it should be explored separately. Whatever variant is responsible could make an exciting drug target.

We thank the reviewer for this comments and valuable and constructive suggestions. Most of the time, BCG was used as a type strain in early research because it does not require cultivation in a biological safety protection third-level laboratory(P3). In our study, we think that the difference in antibiotic susceptibility between the BCG and H37RV strains is attributable to mutations in certain efflux pump genes within the bacteria, resulting in impaired efflux pump function. Behr et al. systematically identified multiple RD regions (such as RD1, RD2, etc.) absent in BCG relative to the Mycobacterium tuberculosis H37Rv strain through genomic comparison for the first time. The absence of these regions constitutes the molecular basis for BCG's attenuated virulence (Behr, M. A., et al. (1999). Comparative genomics of BCG vaccines by whole-genome DNA microarray. Science, 284(5419), 1520-1523.). Although no completely efflux pump genes are missing in BCG in the RD regions, several studies have identified specific gene mutations related to drug efflux and sensitivity, such as, Danilchanka et al. and Ramón-García et al. have indicated that the Rv1410c gene encodes a crucial efflux pump associated with drug resistance and cell envelope integrity in mycobacteria. Within BCG strains, a single nucleotide polymorphism (SNP) exists within the Rv1410c gene, resulting in an altered protein sequence (A202E). This mutation has been demonstrated to reduce the efflux pump's functionality, rendering BCG more susceptible to certain drugs (such as streptomycin and chloramphenicol) than H37Rv (Danilchanka, O., et al. (2008). Identification of a novel multidrug efflux pump of Mycobacterium tuberculosis. Antimicrobial Agents and Chemotherapy, 52(7), 2503-2511; Ramón-García, S., et al. (2011). The Mycobacterium tuberculosis efflux pump Rv1410c confers resistance to a diverse range of antituberculosis drugs. Antimicrobial Agents and Chemotherapy, 55(5), 2294-2297.). We once again express our gratitude to the reviewer for their valuable advice.

2. Figures 2A-J - it was not clear from Communications Biology's instructions to authors that the authors are limited to 2 tables. It would be worth checking with the editors as these figure panels can be much more efficiently presented as tables.

We thank the reviewer for this comments and have added the tables of the synergistic effect of dinactin with rifampicin or isoniazid against drug-resistant M. tuberculosis in manuscript in line 240-242. For figure 2A-G, we have decided to present our results in the form of a line graph.

3. Figure S2 - though the chromatograms are a welcome addition, this data remains unconvincing, particularly for rifampicin, which implies a 40% difference in cell number used for dinactin-treated vs. DMSO-treated samples if correct. As this data is not crucial to the conclusion of the paper it would probably be best to leave it out.

We appreciate the reviewer's comments and have deleted this result in our manuscript.

4. Figure 4A, B, E, I, J - it would be easier for the reader if the time of treatment was in the figure

legend rather than buried in the supplementary

We appreciate the reviewer's comments and advice and we have added the time of treatment in the corresponding legend in line 373-390.

5. Supplementary L62 - says isoniazid although this control has been changed to rifampicin

We appreciate the reviewer's comments and have corrected in supplementary materials in line .

6. Supplementart L192 -the timing here is not clear, 72 hour treatment is clear, 10 minute incubation after resasuzrin is clear, but what what is happening for the 1, 2, and 3 hours? The 10 minutes with resazurin seems very fast, usually incubation for MIC are overnight.

We appreciate the reviewer's comments and for pointing out the incorrect description in the part of "Resazurin reduction assay" in supplementary materials in line 171-182. In this assay, the bacterial cultures were treated with the compounds for 1 h, 2 h, and 3 h respectively, and then the bacterial cultures were taken out at the corresponding time points for subsequent measurements. Resasuzrin, as an indicator for measuring the respiratory chain activity of Mycobacterium tuberculosis, is sufficient for monitoring the respiratory chain activity of Mycobacterium tuberculosis after incubation for 10 minutes. We have added the relative reference into the manuscript in supplementary materials in line 173. (Müller A et al. Daptomycin inhibits cell envelope synthesis by interfering with fluid membrane microdomains. Proc Natl Acad Sci U S A. 2016;113(45):E7077-E7086. doi:10.1073/pnas.1611173113)

7. Supplementary overall - this would benefit from careful proofreading. Isoniazid is again spelled here in two different ways and there are numerous misspellings throughout.

We appreciate the reviewer's comments and apologize for our carelessness and we have detected and corrected these misspellings in our manuscrip.